# Antifungal Polyacetylenic Deoxyglycosides Isolated from Endophytic Fungus *Xylaria* sp. VDL4 Associated with *Vaccinium dunalianum*

**DOI:** 10.3390/jof11030209

**Published:** 2025-03-08

**Authors:** Jiao Yao, Sai Huang, Lingfeng He, Shengyun Wei, Wei Yang, Qiangxin Zhang, Weihua Wang, Xiaoqin Yang, Sida Xie, Yunxian Li, Ping Zhao, Guolei Zhu

**Affiliations:** 1Key Laboratory of State Forestry Administration on Highly-Efficient Utilization of Forestry Biomass Resources in Southwest China, Southwest Forestry University, Kunming 650224, China; jiaoyao@163.com (J.Y.); 17860287601@163.com (S.H.); greennre@163.com (L.H.); 17606998486@163.com (S.W.); yw15096711968@163.com (W.Y.); 15087712145@163.com (Q.Z.); 34016@ztu.edu.cn (W.W.); yangxiaoqin@swfu.edu.cn (X.Y.); dream102035@163.com (S.X.); liyunxian@163.com (Y.L.); 2Yunnan Key Laboratory of Gastrodia and Fungi Symbiotic Biology, Zhaotong University, Zhaotong 657000, China

**Keywords:** *Xylaria* sp., polyacetylenic rhamnosides, antifungal activity, α-glucosidase inhibitory activity, endophytic fungus, *Vaccinium dunalianum*, natural products

## Abstract

One novel C_10_ polyacetylene rhamnoside, 4,6,8-decatriyne-1-*O*-α-L-rhamnopyranoside, named xylariside A (**1**), together with two novel C_10_ polyacetylene quinovopyranosides, 4,6,8-decatriyne-1-*O*-α-D-quinovopyranoside, xylariside B (**2**), and 8*E*-decaene-4,6-diyne-1-*O*-α-D-quinovopyranoside, xylariside C (**3**), were obtained from the solid fermentation of *Xylaria* sp. VDL4, an endophytic fungus isolated from *Vaccinium dunalianum* wight (Ericaceae). Their chemical structures were elucidated through a combination of spectroscopic techniques. The antifungal activities of these compounds were evaluated in vitro against four phytopathogenic fungi (*Fusarium oxysporum*, *Botrytis cinerea*, *Phytophthora capsici*, and *Fusarium solani*). Compound **2** demonstrated significant antifungal activities, with minimum inhibitory concentration (MIC) values ranging from 3.91 to 7.81 μg/mL. Compound **2**’s effectiveness levels were similar to those of the reference drugs thiabendazole and carbendazim (each MIC = 0.98−15.62 μg/mL). Xylariside B (**2**) was further evaluated against *B. cinerea* in vivo. It exhibited remarkable efficacy in both the prevention and treatment of tomato and strawberry gray mold. Molecular docking studies confirmed the antifungal mechanism of compound **2** by revealing its binding interactions with key enzyme targets in *B. cinerea*, thereby supporting the observed in vitro and in vivo results. Additionally, compound **2** showed effective inhibition of α-glucosidase, with IC_50_ values of 5.27 ± 0.0125 μg/mL.

## 1. Introduction

Polyacetylenes, possessing two or more alkynyl functionalities, are a category of secondary metabolites mainly generated by plants and fungi [1,2,3,4]. These privileged chemical structures have garnered tremendous interest for their extensive biological properties, leading to their application in pharmacology, medicinal chemistry, food chemistry, and agricultural protection [5,6]. Their potential in addressing health and disease-related challenges in both animals and humans further underscores their importance. The biosynthesis of polyacetylenes starts with saturated fatty acids, which are generated by the addition of malonyl units to an acyl chain. There are two key ways to form carbon–carbon triple bonds. One is the dehydrogenation of double bonds under the action of NAD or NADP, iron, and molecular oxygen [7]. The other is the elimination reaction of an activated enol carboxylate intermediate in polyketide-derived acetylenic natural products [8]. This leads to the production of three monoacetylenic intermediates: crepenynic acid, stearolic acid, and tariric acid. These intermediates then introduce conjugated alkynes through further unsaturation reactions, shorten the chain length, and introduce functional groups, such as hydroxyl groups, through oxidation reactions. They can also undergo reactions such as esterification, glycosylation, decarboxylation, and sulfur atom addition, ultimately synthesizing polyacetylenes with diverse structures [9]. In plant–pathogen interactions, natural polyacetylenes exhibit notable antifungal activity. For example, falcarinol-type polyacetylenes from carrots (*Daucus carota*) confer protection against *Botrytis cinerea* [10], while celery (*Apium graveolens*) and other Apiaceae plants produce antifungal polyacetylenes [11]. Their antimicrobial properties also contribute to controlling microbial growth in food systems, thereby enhancing food safety and quality. At the cellular level, polyacetylenes disrupt fungal membranes by altering fluidity and permeability, as well as damaging membrane proteins [12]. Additionally, they interfere with intracellular signal transduction pathways, including the mitogen-activated protein kinase (MAPK) and calcium signaling systems, ultimately inhibiting fungal growth and reproduction [5]. Notably, polyacetylenes isolated from *Phoma* fungi suppress the fatty acid synthesis pathway (FASII) to impede fungal proliferation [2,13,14,15]. Certain polyacetylenes demonstrate dual functionality; beyond antimicrobial effects, they exhibit neuroprotective properties such as inhibiting neuronal apoptosis and modulating neurotransmitter levels [16]. These biological activities extend beyond biomedicine. In the food industry, polyacetylenes may prevent nutrient degradation and spoilage during storage, thereby extending shelf life while preserving nutritional quality. Collectively, their diverse bioactivities position polyacetylenes as valuable compounds for healthcare and food safety applications. Recent studies highlighting their inhibitory effects against common plant pathogens further suggest the potential for developing novel plant protection agents. However, some plant-derived polyacetylenes exhibit toxicity. For instance, cicutoxin from the *Cicuta* species causes livestock fatalities [5], and falcarinol in ivy (*Hedera* spp.) triggers contact dermatitis in humans [17]. Importantly, these toxicological properties are largely unrelated to the food-protective functions of polyacetylenes. Furthermore, it is critical to emphasize that, to date, polyacetylenes extracted from endophytic fungi have not demonstrated toxicity to animals or humans.

*Xylaria* sp. VDL4, an endophytic fungus isolated from *Vaccinium dunalianum* leaves (a plant inhabiting ecologically stressed niches), produces bioactive polyacetylenes with demonstrated antifungal and antimicrobial activities that may contribute to host defense mechanisms [18,19,20]. The genus *Xylaria* encompasses both medicinal and edible fungi. Notably, some species form sclerotia within abandoned termite nests; these sclerotia, known as Wulingshen in traditional Chinese medicine, possess therapeutic properties, including immune enhancement, prostatitis treatment, and hematopoiesis promotion [21,22]. As plant endophytes, *Xylaria* species enhance host stress resistance, suggesting their biocontrol potential could reduce chemical pesticide reliance in sustainable agriculture, thereby benefiting both human health and ecosystem preservation. Given *Xylaria*’s documented capacity for synthesizing unique antifungal polyacetylenes, we investigated *Xylaria* sp. VDL4 to identify novel bioactive metabolites for agricultural pathogen control. As a result, one novel polyacetylenic rhamnopyranoside, xylariside A (**1**), as well as two novel polyacetylenic quinovopyranosides, xylariside B (**2**), and xylariside C (**3**) (Figure 1), were yielded from the solid fermented medium. Their structures were elucidated through spectroscopic methodologies and acid hydrolysis in conjunction with GC analysis and comparison with established standards. Their isolates were assessed for their antifungal activities in vitro against four plant pathogenic strains, *Fusarium oxysporum*, *Botrytis cinerea*, *Phytophthora capsici*, and *Fusarium solani*, as well as their inhibitions toward α-glucosidase. Molecular docking studies further confirmed the antifungal mechanism of compound **2** by revealing its binding interactions with key enzyme targets in *B. cinerea*, thereby supporting the observed in vitro and in vivo results. For the significant antifungal activities of compound **2**, it was further evaluated in vivo against *B. cinerea*, which can cause gray mold in tomatoes and strawberries.

## 2. Materials and Methods

### 2.1. General Experimental Procedures

Ultraviolet, infrared, and optical rotation measurements were conducted using a Shimadzu UV2401PC (manufactured by Shimadzu, headquartered in Kyoto, Japan), a Bruker Tensor-27 with KBr pellets, and a JASCO P-1020 polarimeter (manufactured by Bruker, headquartered in Karlsruhe, Germany), respectively. One-dimensional (1D) and two-dimensional (2D) nuclear magnetic resonance (NMR) spectra were obtained utilizing Bruker DRX-500 (manufactured by Bruker, headquartered in Karlsruhe, Germany), with tetramethylsilane (TMS) employed as the internal standard under ambient conditions. High-resolution time-of-flight electrospray ionization mass spectrometry data were obtained using the Agilent G 6230 TOF spectrometer (manufactured by Agilent, headquartered in Santa Clara, California, USA). Column chromatographic (CC) separations were conducted using silica gel (200−300 mesh, Qingdao Marine Chemical Co., Ltd., Qingdao, China), reversed-phase C18 (40−63 μm, Merck, Darmstadt, Germany), and Sephadex LH-20 (20−150 μm, Pharmacia, Uppsala, Sweden). Semi-preparative HPLC separations were conducted utilizing an Agilent 1260 series system equipped with a C18 column (ZORBAX SB, a product of Agilent, headquartered in Santa Clara, California, USA,, 5 μm particle size, 9.4 mm × 250 mm).

### 2.2. Gathering and Characterization of Fungal Material

The endophyte *Xylaria* sp. VDL4 was isolated from the healthy leaf tissues of *Vaccinium dunalianum* collected in Wuding County, Yunnan Province, China, in 2016. After collection, the leaves were strictly disinfected to remove contaminants. The disinfected leaves were cut into 0.5 cm × 0.5 cm pieces and inoculated on a Potato Sucrose Agar (PSA) medium. The inoculated culture medium was placed in the dark at a temperature of 28 °C and a relative humidity of 50–80% for 7 days. When mycelium emerged on the leaf piece edges, the mycelium’s front end was selected and transferred to fresh PSA medium for further growth. After 3–4 generations of purification culture, a pure strain was obtained. The strain’s colony had an uneven, convex-centered appearance, with white mycelium and oval spores. To identify the strain, DNA of VDL4 was extracted and amplified via PCR using fungal universal primers. The PCR products were analyzed by agarose gel electrophoresis, sequenced, and the results were used for a homology search in the NCBI database (https://www.ncbi.nlm.nih.gov/, accessed 18 September 2024). The search showed 100% homology with *Xylaria* sp.; thus, the fungus was identified as *Xylaria* sp. (Appendix A).

Of the four test phytopathogenic fungi, *F. oxysporum*, *B. cinerea*, and *F*. *solani* were obtained from China General Microbiological Culture Collection Center, and *P. capsici* was obtained from Northwest Agriculture and Forestry University research and development center of biorational pesticide.

### 2.3. Fermentation and Isolation

First, take out the endophytic fungus *Xylaria* sp. VDL4 stored in the glycerol–water cryotube. After it thaws naturally, transfer it onto the Potato Sucrose Agar (PSA) medium. Subsequently, place the inoculated medium in the dark at a temperature of 28 °C and a relative humidity of 50–80% for 7 days. Then, the mycelium of this endophytic fungus can be obtained. Next, a hole punch was used to create circular samples with a diameter of 5 mm on the cultivated *Xylaria* Petri dish. Under sterile conditions, the fungus *Xylaria* sp. VDL4 cultured in a PSA medium was inoculated into sterile 500 mL rice fermentation bottles. Each bottle contained 70.0 g of rice and 70.0 mL of water. A total of 15.0 kg of rice was apportioned into about 200 containers and subsequently inoculated. The inoculated samples were then cultured in a constant temperature incubator at 25 °C for 30 days. Following fermentation, the mixture was carefully decanted from the fermentation bottles and transferred to a barrel. Subsequently, the mixture underwent extraction through crushing and soaking in ethyl acetate for 24 h at room temperature. The extraction process was repeated six times, and the resulting filtrates were combined. The combined filtrate was dried and concentrated to obtain approximately the extract (500 g). The extract was adsorbed onto D101 macroporous resin and subsequently eluted with a MeOH-H_2_O gradient (0–100%). The 90% aqueous methanol extracts were concentrated under reduced pressure to produce 120.0 g of residue. This residue was then purified using silica gel column chromatography with a gradient of methylene chloride and methanol [100:1, 30:1, 15:1, 10:1, 5:1, 0:1], resulting in fifteen fractions (A-O). Fraction K (1.2 g) was selected to chromatograph over Rp-C_18_ column eluted with MeOH-H_2_O (gradient from 30:70 to 100:0), providing ten subfractions (K.1–K.10). The subfraction K.6 (35.0 mg) underwent repeated column chromatograph and was subsequently purified using semipreparative HPLC with a mobile phase (45% CH_3_CN-H_2_O, 3.0 mL/min, 210 nm) to give compounds **1** (4.6 mg) and **2** (4.7 mg). Using similar procedures, compound **3** (5.3 mg) was obtained from the K-7 subfraction (15.0 mg).

### 2.4. Spectroscopic Data of New Compounds

Xylariside A (**1**): Light yellow oily solution; ${\left[\alpha \right]}_{D}^{25}$–107.82 (*c* 0.110, MeOH); UV (MeOH) *λ*_max_ (log *ε*) 264.0 (2.95), 207.8 (5.02), 192.6 (3.98) nm; IR (KBr) *v*_max_ 3415, 2933, 2893, 2220, 1710, 1385, 1099, 981 cm^−1^; HRESIMS *m/z* 315.1208 [M + Na]^+^ (calcd for C_16_H_20_O_5_Na, 315.1208); ^1^H NMR (500 MHz, Methanol-*d*_4_) and ^13^C (125 MHz, Methanol-*d*_4_) data, see Table 1.

Xylariside B (**2**): light yellow oily solution; ${\left[\alpha \right]}_{D}^{25}$−146.67 (*c* 0.120, MeOH); UV (MeOH) *λ*_max_ (log *ε*) 207.8 (4.99), 192.6 (3.94) nm; IR (KBr) *v*_max_ 3392, 2930, 2220, 1710, 1382, 1047, 919 cm^−1^; HRESIMS *m/z* 315.1209 [M + Na]^+^ (calcd for C_16_H_20_O_5_Na, 315.1208); ^1^H NMR (500 MHz, Methanol-*d*_4_) and ^13^C (125 MHz, Methanol-*d*_4_) data, see Table 1.

Xylariside C (**3**): colorless oil solution; ${\left[\alpha \right]}_{D}^{25}$−109.40 (*c* 0.100, MeOH); UV (MeOH) *λ*_max_ (lg *ε*) 206.3 (4.36), 210.2 (4.41), 225.8 (3.54) nm, 240.0 (3.64), 251.4 (3.72), 265.6 (3.87), 281.8 (3.76), 308.2 (2.60) nm; IR (KBr) *v*_max_ 3400, 2928, 2231, 1627, 1384, 1046, 950 cm^−1^; HRESIMS *m/z* 317.1365 [M + Na]^+^ (calcd for C_16_H_20_O_5_Na, 317.1364); ^1^H NMR (500 MHz, Methanol-*d*_4_) and ^13^C (125 MHz, Methanol-*d*_4_) data, see Table 1.

### 2.5. Acid Hydrolysis

Each of the above three compounds (**1**–**3**, 1.5 mg) was refluxed with a 5 M trifluoroacetic acid/dioxane mixture (1:1, *v*/*v*, 2.0 mL) for 4 h. Following neutralization with 1 M NaOH, the reaction mixture was filtered. The filtrate was then separated using CHCl_3_ and H_2_O. The aqueous phase was dried to yield a monosaccharide. Subsequently, the sugar residues were dissolved in anhydrous pyridine (1.0 mL) and stirred at 60 °C for 5 min with trimethylchlorosilane (TMCS). The solvent was removed under a stream of N_2_, and the resulting mixture was extracted with diethyl ether. The extract was analyzed by GC (SGE AC-10 quartz capillary column, 0.25 μm, 30 m × 0.32 mm; temperature program: 180–280 °C at 3 °C/min; carrier gas, N_2_, 2 mL/min; injector and detector temperatures: 250 °C; injection volume: 2 μL; split ratio: 1/50) [23,24]. The monosaccharide composition of each compound was identified by comparing the retention times with those of standard sugars.

### 2.6. Activity Determination of Compounds (***1**–**3***)

#### 2.6.1. In Vitro Antifungal Activity Assay

The antifungal activities of compounds (**1**–**3**) were evaluated in vitro using the microbroth dilution method in 96-well plates with PDA medium. Thiabendazole and carbendazim, provided by Aladdin Chemical Co., Ltd. (Shanghai, China), were used as positive controls, while an equivalent concentration of dimethyl sulfoxide (DMSO) solution served as the negative control. The fungi were cultured in PD broth at 28 ± 0.5 °C for 48 h, and then spore suspensions were prepared by dilution in PDB broth to a concentration of approximately 1 × 10^6^ CFU/mL. The dilution experiments were conducted using 96-well plates. In the first column of each row, 50 μL of stock solution (mother liquor) and 100 μL of PDA medium were added to ensure the stock solution remained uncontaminated. In the second column, 100 μL of stock solution was added. From the third to the twelfth columns, 50 μL of medium was added. Then, 50 μL of the stock solution from the second column was transferred to the third column, thoroughly mixed, and sequentially transferred to the fourth column, continuing this process until the eleventh column. The 50 μL removed from the eleventh column was discarded into a waste container. Finally, 100 μL of fungal suspension was added to each well from the second to the twelfth columns, with the twelfth column serving as a control to confirm that the fungal suspension was uncontaminated. The compound concentrations started at an initial value of 250 μg/mL and were serially diluted nine times, reaching a minimum concentration of 0.49 μg/mL. Each compound was tested in duplicate. After incubation at 28 ± 0.5 °C for 48 h, the growth of pathogenic fungi was observed (Appendix A). The minimum inhibitory concentration (MIC) was defined as the lowest concentration of the test compound in each well at which no microbial growth was observed [25].

#### 2.6.2. In Vivo Antifungal Activity Assay

The control effects of compounds **1**, **2**, and **3** on gray mold of fruits and vegetables were evaluated. In this experiment, we carefully selected tomato and strawberry fruits of uniform size. These fruits were then randomly divided into six different groups, including three groups of test compounds, a blank control group, and two positive control groups. Each group had 3 replicate subgroups, with 3 fruits in each subgroup. Thus, 18 fruits were required for a single experiment. To further verify the stability, we carried out three independent replicate experiments, using a total of 54 fruits. Each fruit was punctured on both sides of the equator with a sterile stainless-steel needle to create two symmetrical and uniform wounds. Each wound was inoculated with 20.0 μL of *Botrytis cinerea* spore suspension at a concentration of 1 × 10^4^ spores/mL and then air-dried at room temperature. Two hours after inoculation, 20.0 μL of compounds **1**, **2**, and **3** (each at a concentration of 250 μg/mL) were applied to the fruits in each group, respectively. After air-drying at room temperature, the fruits were placed in a foam box, sealed in a polyethylene bag, and maintained at a relative humidity of 90–95% with a wet gauze. Then, they were stored at 25 °C until the onset of the disease. The disease incidence was defined as the percentage of infected lesions showing signs of decay. On the 4th day after inoculation, the lesion diameters of tomatoes and strawberries were measured using the cross-method with a vernier caliper. The formula is Lesion diameters (cm) = (∑ Lesion diameters of infected fruits)/(Total number of fruits in each treatment) [26,27].

In this study, statistical analyses were performed using Origin 2021 and SPSS 19.0 software. To assess the effects of the treatment measures, one-way analysis of variance (ANOVA) was applied. Significant differences among experimental groups were further examined using Duncan’s new multiple range test. The *p*-value derived from comparisons of the means of three replicate experiments was used as the significance criterion, with a threshold of *p* < 0.05 indicating statistically significant differences.

#### 2.6.3. α-Glucosidase Inhibitory Activity

The enzyme solution was prepared with α-glucosidase at a concentration of 0.5 units per milliliter (U/mL). Compounds **1**−**3** were assessed for their α-glucosidase inhibitory activity utilizing acarbose as the positive control. Different concentrations (Appendix A) of the purified compounds (20.0 μL) were mixed with the enzyme solution (50.0 μL) and incubated at 37 °C for 15 min. The reaction was initiated by adding p-nitrophenyl-α-D-glucopyranoside (40.0 μL, 5 mM). After incubation for 25 min at 37 °C, we added 0.1 M sodium carbonate (Na_2_CO_3_) (50 μL). Absorbance (A) was measured at 405 nm. The inhibitory activity was expressed as a percentage of enzyme activity inhibition and calculated using the formula: I = [1 − (A_sample determination group_ − A_sample control group_)/(A_blank group_ − A_blank control group_)] × 100%. Here, A_sample determination group_ denotes the absorbance of the sample solution and enzyme, A_sample control group_ refers to the absorbance of the sample solution and PBS, A_blank group_ indicates the absorbance of PBS substituting for the sample and enzyme, and A_blank control group_ signifies the absorbance of PBS substituting for the sample. The 50% inhibitory concentration (IC_50_) was derived using the Logistic function from the Growth/Sigmoidal category in Origin 2021, employing the Levenberg–Marquardt iterative optimization algorithm [28,29].

The water used in the experiment was ultrapure water. The α-glucosidase extracted from *Saccharomyces cerevisiae*, PBS phosphate buffer, p-nitrophenyl-α-D-glucopyranoside (batch number S10137), and acarbose hydrate (98%) were purchased from Shanghai Yuanye Biotechnology Co., Ltd. (Shanghai, China). DMSO was obtained from Beijing Solarbio Science & Technology Co., Ltd. (Beijing, China). The SpectraMax 190 microplate reader used in this experiment was produced by Shanghai Molecular Devices Co., Ltd. (Shanghai, China).

#### 2.6.4. Molecular Docking

Molecular docking was performed using AutoDock Vina (version 1.2.0) to predict the binding interactions between compound **2** and the target enzyme. The protein structure was prepared using AutoDock Tools, including hydrogen addition and charge assignment, while the ligand structure was energy-minimized using Avogadro (version 1.2.0). Visualization of the docking results was conducted using PyMOL (version 2.5.0) and LigPlot+ (version 2.2) to analyze the binding poses and interactions.

## 3. Results

### 3.1. Structure Elucidation of Compounds ***1**–**3***

Xylariside A (**1**) was obtained as a light-yellow oil, and its molecular formula was determined to be C_16_H_20_O_5_ based on the interpretation of its HRESIMS peak at *m/z* 315.1208 [M + Na]^+^ (calcd. 315.1207), mandating the presence of seven indices of hydrogen deficiency. The IR spectrum exhibited an absorption band at 3415 cm^−1^, indicating the presence of characteristic hydroxyl groups, as well as conjugated acetylene functionalities at 2220 cm^−1^. In the ^1^H NMR spectrum of **1** (Table 1), two methyl signals at *δ*_H_ 1.95 (3H, s), 1.25 (d, *J* = 6.2 Hz, 3H), and one anomeric proton at *δ*_H_ 4.64 (d, *J* = 1.5 Hz, 1H) were observed. With the aid of DEPT and HSQC spectra, 16 carbon resonant signals observed in ^13^C NMR spectrum could be classify as two methyls at *δ*_C_ 3.7 (C-10), 18.0 (C-1′), three methenes (including one oxygenated at *δ*_C_ 66.4), five methines (including one anomeric at *δ*_C_ 101.6), and six acetylenic quaternary carbons at *δ*_C_ 79.1, 76.2, 66.7, 65.2, 61.3, and 60.1. These aforementioned spectroscopic data suggested **1** was a C_10_ polyacetylenic monoglycoside, and the aglycone resembled the reported carthamoside A_2_, except an additional alkynyl appeared in **1** rather than a double bond at C-8 and C-9 in the known compound [24,30]. In comparison with the corresponding chemical shifts of carthamoside A_2_, an obvious upfield shift was observed for the terminal methyl at C-10 (Δ*δ*_C_ 14.8), suggesting that the alkynyl was located at C-8 and C-9. This was supported by the HMBC correlations from Me-10 to C-7, C-8, and C-9. Then, the aglycone was absolutely determined after detail speculation of the 2D NMR (Figure 2).

The sugar unit of **1** was determined to be L-rhamnopyranosyl through acid hydrolysis, as well as GC analysis and comparison with validated reference samples. Its location at C-1 of the aglycone was disclosed via HMBC correlation between H-1′ (*δ*_H_ 4.64) and C-1 (*δ*_C_ 66.4), while the α-orientation of the L-rhamnopyranosyl was deduced by ^13^C NMR chemical shift of C-5′ (*δ*_C_ 69.9), and the ^13^C-^1^H coupling constant between H-1′ and C-1′ (^1^*J*_CH_ = 170 Hz) (Appendix A) [23,31]. Therefore, the structure of **1** was elucidated as 4,6,8-decatriyne-1-*O*-α-L-rhamnopyranoside, namely xylariside A (**1**).

Xylariside B (**2**), a light, oily solution, has the same molecular formula as xylariside A (**1**) based on the HRESIMS peak at *m/z* 315.1209 [M + Na]^+^ (calcd. 315.1208). Detailed inspection of their ^1^H and ^13^C NMR data (Table 1) suggested that they possess the same aglycone, and a distinct 6-deoxy sugar was replaced at the C-1 position in compound **2**. Compared to the corresponding chemical shifts observed in the ^13^C NMR spectrum of the sugar moiety, we observed upfield shifts for C-1′/C-5′ (Δ*δ*_C_ 1.5 and 1.0, respectively), and noticeable downfield shifts were detected for C-2′/C-3′/C-4′ (Δ*δ*_C_ 1.5, 2.4, and 3.5, sequentially). Incorporation with the coupling constants of H-1′ with H-2′ (*J* = 3.8 Hz), H-2′ with H-3′ (*J* = 9.6 Hz), H-3′ with H-4′ (*J* = 9.6 Hz), H-4′ with H-5′ (*J* = 9.5 Hz), and H-5′ with Me-6′ (*J* = 6.2 Hz) observed in the ^1^H NMR and HSQC spectra, as well as ROESY cross-peaks between H-2′ and H-4′, and H-3′ and Me-5′ (Figure 3) indicated that the sugar unit was a 6-dexy sugar, which was further determined as D-quinovose moiety after acid hydrolysis together with GC analysis and comparison with authentic samples [32,33,34]. The α-orientation of the D-quinovopyranosyl was identified via the small coupling constants (*J* = 3.8 Hz) of H-1′ with H-2′ detected from the ^1^H-^1^H NMR spectrum. As illustrated in Figure 3, the structure was validated through its 2D NMR data. Therefore, the structure was elucidated as 4,6,8-decatriyne-1-*O*-α-D-quinovopyranoside (**2**) and named xylariside B.

Xylariside C (**3**), a colorless oil, possesses a molecular formula of C_16_H_20_O_5_, with six indices of hydrogen deficiency, determined by its quasi-molecular ion at *m/z* 315.1209 [M + Na]^+^ (calcd. 315.1208) in the HRESIMS spectrum. The molecular formula of **3**, with six indices of hydrogen deficiency, showed two mass units more than **2**, lacking one degree of unsaturation. Incorporating with their 1D NMR spectra (Table 1), this difference can be illustrated by one olefinic bond [*δ*_C_ 110.9, 144.1; *δ*_H_ 5.54 (1H, d, *J* = 15.9 Hz), 6.27 (1H, td, *J* = 15.8, 6.8 Hz)] appearing in **3** instead of the acetylenic carbon at C-8 and C-9 in **2**. This was further supported by the downfield shift of C-7 (Δ*δ*_C_ 13.5), HMBC cross-peaks from Me-10 to C-9, C-8, and C-7 (*δ*_C_ 74.7), and COSY correlations of Me-10/C-9/C-8 (Figure 2). Its configuration of Δ^8^ was assigned as *E* based on the large coupling constant of H-8/H-9 (^3^*J*_H8-H9_ = 15.9 Hz), which was supported by the ROESY correlation between H-8 and Me-10 (Figure 3). Moreover, the remaining signals in the structure of the aglycon were absolutely identical with **2**, and the sugar moiety was determined to be the same as that of **2** using the same method. Thus, the structure was elucidated as 8*E*-decaene-4,6-diyne-1-*O*-α-D-quinovopyranoside and named xylariside C (**3**).

### 3.2. Results of the In Vitro Antifungal Activity Assay

All of the polyacetylenic rhamnosides (**1**–**3**) were evaluated in vitro toward four plant pathogenic fungi, *F. oxysporum*, *B. cinerea*, *P. capsici*, and *F. solani*, with two commercial fungicides, thiabendazole and carbendazim, as the positive controls. As a result, their antifungal effects were observed to vary on the above pathogens, with MIC values ranging from 3.91 to 31.25 μg/mL (Table 2), in which xylariside B (**2**) had the strongest inhibitory activities toward the test pathogenic fungi. Especially, its inhibitions were superior to the positive control carbendazim (MIC = 15.62 μg/mL) for *F. oxysporum* and were superior to the positive control thiabendazole (MIC = 15.62 μg/mL) for *P. capsici.* In conjunction with their inhibitory activities and structural characteristics, it can be concluded that a linear molecular linkage featuring an α-D-quinovopyranoside motif may serve as the pharmacophore.

### 3.3. Results of the In Vivo Antifungal Activity Assay

The plant disease fungus *B. cinerea* is acknowledged as a highly detrimental pathogen causing gray mold in an extensive variety of vegetables, fruits, and ornamental plants [26,35]. To prevent the occurrence of tomato and strawberry gray mold disease, further evaluations to assess the effectiveness of xylariside A−C (**1−3**) toward *B. cinerea* were carried in vivo based on their antifungal performance observed in laboratory tests. As illustrated in Figure 4 and Figure 5, compound **2** exhibits a substantial protective effect at a concentration of 250.0 μg/mL. The ulcer area in the protected group was significantly reduced compared to the blank control group, equivalent to the positive control (*p* < 0.05).

### 3.4. Molecular Docking Analysis of Compound ***2***

In the intricate life cycle and pathogenesis of *Botrytis cinerea*, Cytochrome P450 51 (CYP51) and Chitin Synthase (CHS) emerge as pivotal molecular targets. CYP51, a key enzyme in the ergosterol biosynthesis pathway, governs fungal membrane integrity and is intrinsically linked to drug resistance and virulence. Functional disruption of CYP51 not only impairs pathogen proliferation but also attenuates its infectivity [36]. Conversely, CHS mediates chitin synthesis critical for maintaining cell wall rigidity and facilitating hyphal morphogenesis-processes essential for both structural stability and host invasion [37]. Molecular docking, a sophisticated computational approach in structural biology, enables the precise prediction of binding modes and affinities between bioactive compounds and target proteins [38]. This technique provides mechanistic insights into the structural basis of antimicrobial activity, complementing experimental observations. In this study, we employed a dual-targeting strategy focusing on molecular docking of compound **2** with CYP51 and CHS. The docking results (Figure 6) revealed dual-target engagement of compound **2** with CYP51 and CHS, exhibiting binding energies of −6.2 and −5.1 kcal/mol, respectively. For CYP51, compound **2** formed an extensive hydrogen-bond network with key residues (Lys-56, Asp-53, Phe-57, Thr-49, Asp-60, and Val-37) and hydrophobic contacts with eight residues, including Asp-53 (Figure 6B,C). Notably, Asp-53, Phe-57, and Asp-60 synergistically stabilized the CYP51–compound complex through both polar and nonpolar interactions. In CHS docking, hydrogen bonding with multiple amino acid residues, including Asn-70 and Gln-71 (Figure 6E), was complemented by hydrophobic interactions involving seven residues (Figure 6F). Crucially, the C-3′-, C-4′-, and C-6′-hydroxyl groups of compound **2** consistently participated in hydrogen bonding across both targets, identifying these moieties as essential pharmacophores for antifungal activity.

### 3.5. Results of α-Glucosidase Inhibitory Activity

Diabetes Mellitus (DM), a metabolic disorder characterized by elevated blood glucose levels, has emerged as a major global health concern, precipitating long-term complications and incurring substantial economic burdens. α-glucosidase, a class of enzymes that can hydrolyze glucosidic bonds, plays a crucial role in the body’s carbohydrate metabolism. Under normal circumstances, it acts on glycosides containing glucose. Substances like maltose and glucose are within its scope of action. In *Saccharomyces cerevisiae* cells, the distribution of α-glucosidase within the cell is specific. Some α-glucosidases are present in the cytoplasm and are involved in the metabolic process of intracellular sugars. Others are associated with the cell membrane and play a role in the process of the cell taking up extracellular sugars and the initial hydrolysis of these sugars. In the context of diabetes, the inhibition of α-glucosidase is pivotal in diabetes treatment [39,40]. Accordingly, the α-glucosidase inhibitory activities of isolates (**1−3**) were assessed (Table 3). Notably, compound **2** demonstrated a remarkable inhibitory effect on α-glucosidase, with an IC_50_ value of 5.27 ± 0.0125 μg/mL.

## 4. Discussion

To date, polyacetylenic compounds have been successfully isolated from various biomasses such as plants, animals, fungi, and sponges. However, among these sources of isolation, obtaining natural polyacetylenic compounds from plants is relatively common. In sharp contrast, reports on the presence of polyacetylenic compounds in endophytic fungi are scarce [41,42]. For the first time, we report the identification of novel natural products from *Xylaria* endophytic fungi. This discovery significantly enhances our understanding of the secondary metabolite diversity in *Xylaria* and paves new avenues for research in this field. In terms of the activity against agricultural pathogenic bacteria, the results of this study are corroborated by multiple similar studies. Reference [10] indicates that certain polyacetylenic compounds isolated from plants can inhibit plant pathogenic bacteria. In this study, compounds **1**–**3** isolated from *Xylaria* sp. VDL4 exhibited varying degrees of antifungal activity against *Fusarium oxysporum*, *Botrytis cinerea*, *Phytophthora capsici*, and *Fusarium solani*. Notably, compound **2** demonstrated particularly significant inhibitory effects, and its inhibitory activity against some pathogens was superior to that of the positive control agents. To further explore the underlying mechanism of compound **2**’s antifungal activity, molecular docking experiments were conducted. The results showed that compound **2** could stably bind to key proteins in these pathogenic fungi, such as those related to cell wall synthesis and energy metabolism pathways. This stable binding mode provided a molecular-level explanation for its strong inhibitory effect, validating the antifungal activity of compound **2** from a structural perspective. This suggests that not only do plant-derived polyacetylenic compounds possess antifungal properties, but also polyacetylenic substances produced by endophytic fungi hold great potential in the control of agricultural pathogenic bacteria, providing new ideas and resources for the development of novel biopesticides.

In the field of α-glucosidase inhibitory activity research, previous studies have shown that some natural products can inhibit the activity of this enzyme, thus playing a positive role in the treatment of diabetes [40]. In this study, compound **2** showed strong inhibitory activity against α-glucosidase, with an IC_50_ value of 5.27 ± 0.0125 μg/mL. This result echoes previous research and further enriches the variety of natural products with α-glucosidase inhibitory activity, offering potential lead compounds for the development of new anti-diabetes drugs. From the perspective of the structure–activity relationship, a comparison between compounds **1** and **2** reveals that they have similar structures but different activities, and the main difference lies in the sugar moiety. This finding is consistent with the research results in reference [31] regarding the influence of the sugar moiety on the biological activity of compounds, indicating that different types of deoxy sugar moieties have a crucial impact on the antibacterial and hypoglycemic activities of compounds. A linear molecular linkage characterized by an α-D-quinovopyranoside motif may serve as the pharmacophore, which provides an important clue for in-depth research on the structure–activity relationship of polyacetylenic compounds and also lays a foundation for subsequent structure-based drug design and optimization. From an ecological perspective, these novel bioactive polyacetylenic natural products in this study are highly likely to have played a positive role in the process of the host plant *Vaccinium dunalianum* resisting biotic stress and adapting to the environment. Similarly, reference [22] mentions that secondary metabolites produced by endophytic fungi can help host plants enhance their resistance to biotic and abiotic stresses. The results of this study further support this view, indicating that there is a close symbiotic relationship between endophytic fungi and host plants, and the polyacetylenic compounds produced by endophytic fungi may be an important part of the host plant’s defense mechanism in this symbiotic relationship. Regarding application prospects, the compounds in this study are expected to serve as fungus-derived lead compounds, providing references for the development of new agricultural or pharmaceutical preparations. This is in line with the current development trend in the agricultural and medical fields, which seek green, efficient, and low-toxicity new drugs [1].

## 5. Conclusions

In conclusion, three novel polyacetylenic glucosides (**1**–**3**) were successfully isolated from the solid fermentation of *Xylaria* sp. VDL4, an endophytic fungus originating from the leaves of *V. dunalianum* Wight. All three structures feature a linear aglycone with a desoxy-sugar unit (rhamnose or quinovose), which represents the first example of such linear aglycone in conjunction with desoxy-sugar. These isolates were assessed in vitro against four phytopathogenic fungi (*F. oxysporum*, *B. cinerea*, *P. capsici*, and *F. solani*) and the inhibitory activities of α-glucosaccharase. Compound **2** showed significant inhibition comparable with the positive control (acarbose). Notably, all of them exhibited varying degrees of biological activity in the inhibition of plant pathogenic fungi with MICs of 3.91–31.25 μg/mL, comparable or better than the positive controls (MICs of 0.98–15.62 μg/mL). Among the isolates, xylariside B (**2**) displayed the strongest inhibitory activities against phytopathogens, otherwise exhibiting superior activities (MIC = 7.81 μg/mL) than the positive control carbendazim toward *F. oxysporum* (MIC = 15.62 μg/mL), and thiabendazole toward *P. capsici* (MIC = 15.62 μg/mL). Moreover, compound **2** exhibited substantial protective and therapeutic benefits against gray mold in both tomatoes and strawberries in vivo. Molecular docking experiments further validated the antifungal activity of compound **2** by revealing its stable binding with key proteins in the target fungi, providing a molecular-level explanation for its inhibitory effect. It is reasonable to suggest that the presence of these biologically active compounds, especially their antifungal activity against phytopathogens, could help protect the host plant *V. dunalianum* from threats posed by parasites, insects, and pathogens. Additional research is needed to clarify the mechanisms behind their antifungal effects and to investigate potential synthetic routes for these bioactive compounds.

## Figures and Tables

**Figure 1 jof-11-00209-f001:**
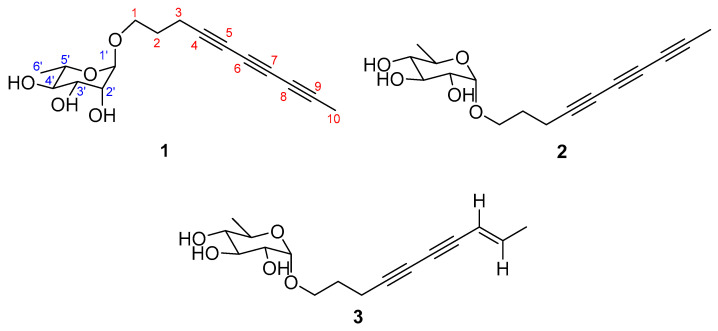
Chemical structures of compounds **1**–**3**.

**Figure 2 jof-11-00209-f002:**
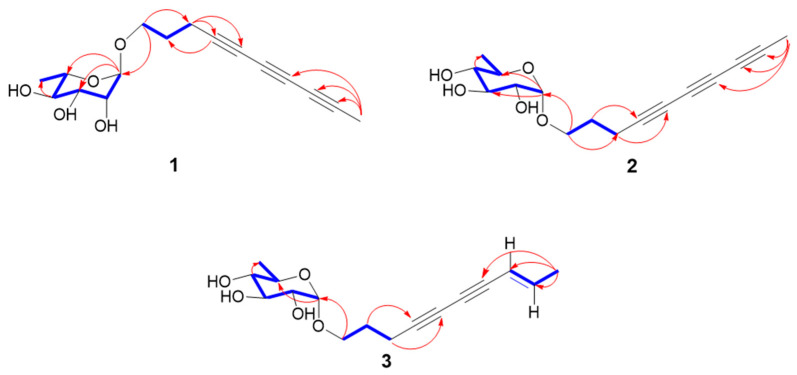
Key HMBC (red arrows) and ^1^H-^1^H COSY correlations (blue bold lines) of **1**–**3**.

**Figure 3 jof-11-00209-f003:**
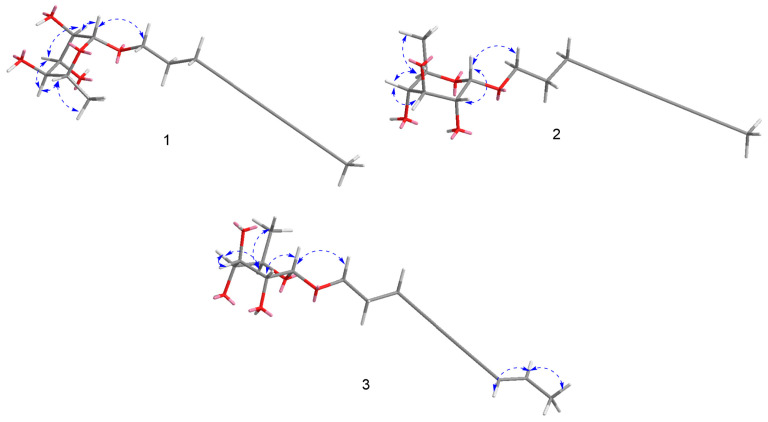
Key ROESY correlations (blue double arrows) of **1**–**3**.

**Figure 4 jof-11-00209-f004:**
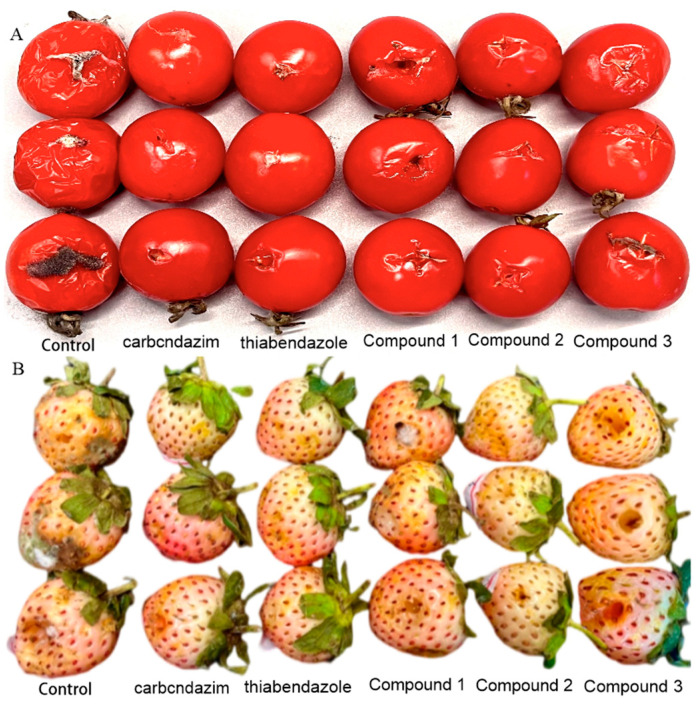
In vivo activities against *B. cinerea* of compounds (**1−3**); (**A**) Bacteriostatic effect of gray mold of tomato; (**B**) bacteriostatic effect of gray mold of strawberry.

**Figure 5 jof-11-00209-f005:**
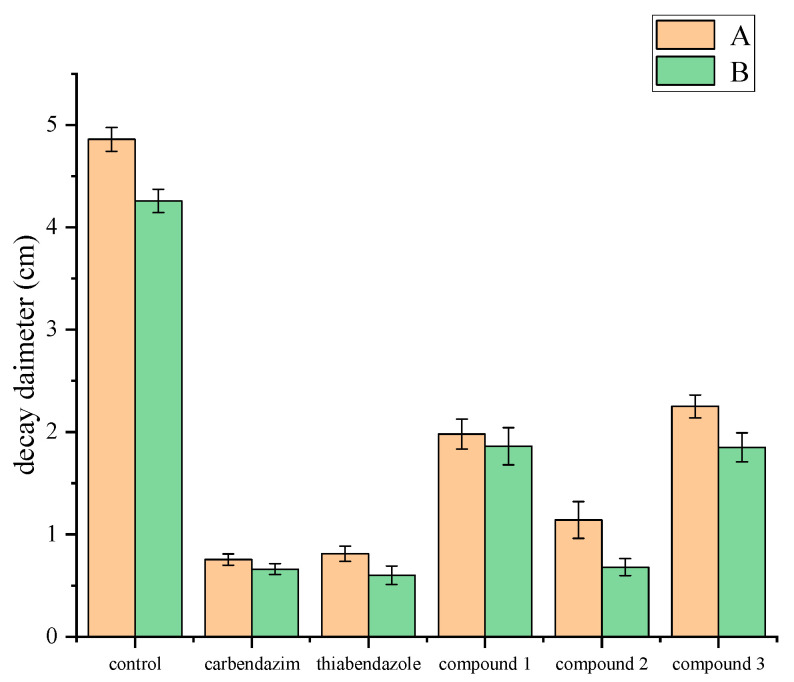
In vivo activities against *B. cinerea* of compounds **1**−**3** at 250.0 μg/mL. (A) Bacteriostatic effect of gray mold of tomato; (B) bacteriostatic effect of gray mold of strawberry; thiabendazole and carbendazim were used as positive controls at a concentration of 250 μg/mL; the vertical bar represents standard deviation of the mean (*n* = 3); the differences were significant, with all *P*-values less than 0.05.

**Figure 6 jof-11-00209-f006:**
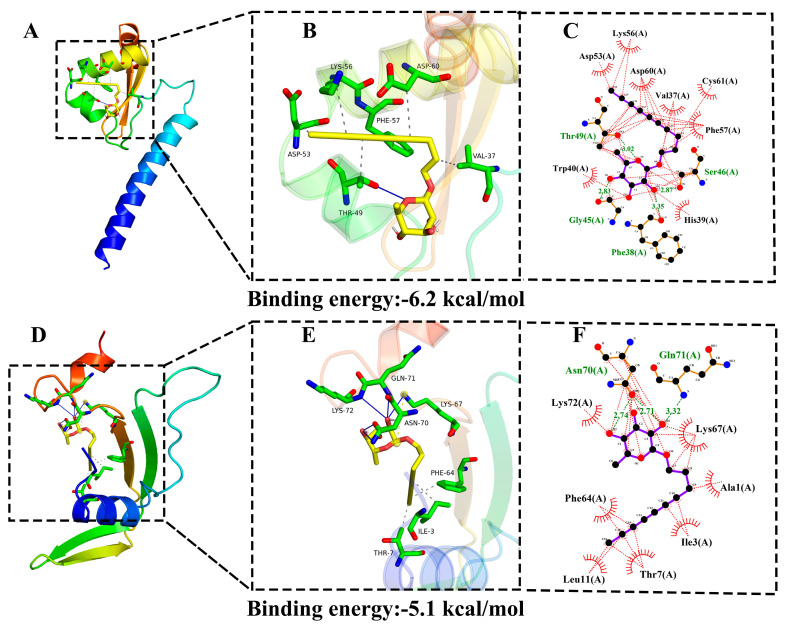
Molecular docking analysis between compound **2** with CYP51 (**A**–**C**) and CHS (**D**–**F**); the green color represents the amino acid residues that are involved in the formation of hydrogen bonds with compound **2**; the gray or green dotted lines represent the hydrogen bond interactions; the red crescent symbols represent the amino acid residues that are involved in the hydrophobic interaction with compound **2**; the red dotted lines represent the hydrophobic interactions.

**Table 1 jof-11-00209-t001:** ^1^H NMR (500 MHz) and ^13^C NMR (125 MHz) data of compounds **1**–**3** in methanol-*d*_4_ (*δ* in ppm, *J* in Hz).

Position	1		2		3	
	*δ*_H_ (*J* in Hz)	*δ* _C_	*δ*_H_ (*J* in Hz)	*δ* _C_	*δ*_H_ (*J* in Hz)	*δ* _C_
1	3.44 m	66.4 t	3.48 d (9.9)	66.6 t	3.81 td (9.8, 7.9, 5.0)	67.0 t
	3.77 d (3.5, 1.75)		3.80 td (9.9, 8.0, 4.9)		3.48 td (9.8, 5.6)	
2	1.27 d (2.7)	29.1 t	1.88 m	29.2 t	1.87 td (13.2, 6.8, 1.7)	29.4 t
	1.78 tdd (13.7, 7.0, 5.1)		1.79 m		1.80 m	
3	2.42 td (6.9, 2.4)	16.7 t	2.48 td (7.0, 4.0)	16.7 t	2.48 td (7.0, 3.8)	16.9 t
4	-	79.1 s	-	79.2 s	-	83.2 s
5	-	61.3 s	-	60.1 s	-	73.4 s
6	-	66.7 s	-	66.9 s	-	66.7 s
7	-	65.2 s	-	65.2 s	-	74.7 s
8	-	60.1 s	-	61.2 s	5.54 d (15.8)	110.9 d
9	-	76.2 s	-	76.2 s	6.27 td (15.8, 6.8)	144.1 d
10	1.95 s	3.7 q	1.95 s	3.1 q	1.79 dd (6.8, 1.8)	18.7 q
1′	4.64 d (1.75)	101.6 d	4.70 d (3.8)	100.1 d	4.70 d (3.8)	100.1 d
2′	3.77 dd (3.5, 1.75)	72.3 d	3.39 dd (9.6, 3.8)	73.8 d	3.38 dd (9.4, 3.8)	73.8 d
3′	3.61 dd (9.5, 3.5)	72.4 d	3.56 br t (9.6)	74.8 d	3.57 br t (9, 4)	74.9 d
4′	3.34 t (9.5)	73.9 d	2.97 t (9.5)	77.4 d	2.97 t (9.45)	77.5 d
5′	3.56 dd (9.5, 6.2)	69.9 d	3.64 dd (9.5, 6.2)	68.9 d	3.65 dd (9.45, 6.2)	68.9 d
6′	1.25 d (6.2)	18.0 q	1.23 d (6.2)	18.1 q	1.23 d (6.2)	18.1 q

**Table 2 jof-11-00209-t002:** Inhibitory effects of compounds **1**−**3** on phytopathogenic fungi.

Compound	MIC ^a^ (μg/mL)			
	Phytopathogenic Fungi			
	*F. oxysporum*	*B. cinerea*	*P. capsici*	*F. solani*
**1**	31.25	31.25	15.62	15.62
**2**	7.81	7.81	7.81	3.91
**3**	15.62	31.25	15.62	7.81
thiabendazole ^b^	3.91	3.91	15.62	0.98
carbendazim ^b^	15.62	3.91	7.81	0.98

^a^ Minimum Inhibitory Concentration; ^b^ thiabendazole and carbendazimab were used as fungal positive controls.

**Table 3 jof-11-00209-t003:** α-glucosidase assay results of compounds **1**−**3**.

Compound	IC_50_ ^a^ (μg/mL)
	α-Glucosidase Inhibitory Activity
**1**	47.69 ± 0.1587
**2**	5.27 ± 0.0125
**3**	15.78 ± 0.0880
Acarbose	0.03 ± 0.0124

^a^ IC_50_ values represent the means ± SD of three parallel measurements.

## Data Availability

Data are contained within the article and Appendix A.

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
