# Peer review of "Antifungal Polyacetylenic Deoxyglycosides Isolated from Endophytic Fungus Xylaria sp. VDL4 Associated with Vaccinium dunalianum"

_jof, 2025, doi:10.3390/jof11030209_

Round 1

Reviewer 1 Report

Review of Manuscript jof-3444661

“Antifungal Polyacetylenic Deoxyglycosides Isolated from Endophytic Fungus Xylaria sp. VDL4 Associated with 3 Vaccinium dunalianum”. The manuscript is of interest due to characterization of novel compounds produced by Xylaria. In this sense the authors need to improve the discussion and conclusion of their findings. There is not literature comparison. Additionally, some particular issues should be described to improve this manuscript.

 1.    In abstract the authors mentioned Phytophthora capsica the correct name is Phytophthora capsici.

 2.    In section 2.2 Gathering and Characterization of Fungal Material, the authors need to added methodology including the isolation, morphological and molecular characterization of of Xylaria sp. VDL4.

 3.    In section 2.3 Fermentation and Isolation, the authors must mention the propagation conditions of Xylaria in PSA medium, also besides the fermentation conditions in solid medium (Temperature, humidity, etc.).

4.    In line 81, the authors mentioned… then a total extract (500 g) was obtained after extracting with equal volume ethyl acetate for 6 times… Is this total extract a sample of the 15 kg taken aleatorily from the end of fermentation in solid substrate and processed with equal volume of ethyl acetate? On the contrary it would be understood that the 15 kg were processed with an equal volume of ethyl acetate obtaining a 500 g extract. Please explain.

 5.    In section 2.6.2 In Vivo Antifungal Activity Assay… The authors describe the methodology for the bacterium Staphylococcus aureus not for the pathogenic fungi (F. oxysporum, B. cinerea, P. capsici, and F. solani).

6.    In line 244 the authors mentioned… Specially, its inhibitions were superior to the positive 244 control carbendazim (MIC = 15.62 μg/ml) for F. oxysporum and thiabendazole (MIC = 25.62 245 μg/ml) for P. capsici… But the thiabendazole for P. capsici in Table 2 is 15.62 μg/ml not 25.62 μg/ml.

 7.    . Correct the name of the fungi Phytophthora capsica in table 2.

No further comments

Author Response

To reviewer #1:

Major comments:

“Antifungal Polyacetylenic Deoxyglycosides Isolated from Endophytic Fungus Xylaria sp. VDL4 Associated with 3 Vaccinium dunalianum”. The manuscript is of interest due to characterization of novel compounds produced by Xylaria. In this sense the authors need to improve the discussion and conclusion of their findings. There is not literature comparison. Additionally, some particular issues should be described to improve this manuscript.

Response: We're extremely grateful for your pointing this out. We fully concur with this. Accordingly, we've added a discussion section to the revised manuscript, delving deeper into the research findings and further refining the relevant content. You can locate this newly added part on page 13, section 4, line 356 of the revised manuscript.

  1. In abstract the authors mentioned Phytophthora capsicathe correct name is Phytophthora capsici.

Response: Thank you for pointing this out. I'm sorry for such a mistake. In the manuscript, we have already corrected the misnomer of Phytophthora capsica based on your advice, which can be found on the first page of the article, in line 21 of the abstract section.

  1. In section 2.2 Gathering and Characterization of Fungal Material, the authors need to added methodology including the isolation, morphological and molecular characterization of Xylaria sp.

Response: Thanks to the reviewer for their professional suggestions. According to these suggestions, we have supplemented the separation methods, morphological features, and molecular characterization of Xylaria sp. VDL4 at the corresponding positions in the manuscript. Modify as follows: The endophyte Xylaria sp. VDL4 was isolated from healthy leaf tissues of V. dunalianum collected in Wuding County, Yunnan Province, China in 2016. Post-collection, the leaves underwent strict disinfection to remove contaminants. The disinfected leaves were cut into 0.5cm × 0.5cm pieces and inoculated on Potato-Sucrose Agar (PSA) medium. The inoculated medium was incubated at 28°C in dark for about 7 days. When mycelium emerged on the leaf-piece edges, the mycelium's front end was selected and transferred to fresh PSA medium for further growth. After 3-4 generations of purification culture, a pure strain was obtained. The strain's colony had an uneven, convex centered appearance, with white mycelium and oval spores. To identify the strain, DNA of VDL4 was extracted and amplified via PCR using fungal universal primers. The PCR products were analyzed by agarose gel electrophoresis, sequenced, and the results were used for a homology search in the NCBI database (blaster.ncbi.nlm.nih.gov). The search showed 100% homology with Xylaria sp., thus identifying the fungus was identified as Xylaria sp. In the revised manuscript, this modification is located on line 80 of Section 2.2, page 2.

  1. In section 2.3 Fermentation and Isolation, the authors must mention the propagation conditions of Xylariain PSA medium, also besides the fermentation conditions in solid medium (Temperature, humidity, etc.).

Response: Thank you for pointing this out. We agree with this comment. Therefore, we have modified the propagation conditions of the endophytic fungus Xylaria sp. VDL4 on the PSA medium to: "First, take out the endophytic fungus Xylaria sp. VDL4 stored in the glycerol-water cryotube. After it thaws naturally, transfer it onto the Potato Sucrose Agar (PSA) medium. Subsequently, place the inoculated medium in the dark at a temperature of 28°C and a relative humidity of 50%-80% for 7 days." This modification can be found on line 106 of Section 2.3, page 3 of the revised manuscript.

  1. In line 81, the authors mentioned… then a total extract (500 g) was obtained after extracting with equal volume ethyl acetate for 6 times… Is this total extract a sample of the 15 kg taken aleatorily from the end of fermentation in solid substrate and processed with equal volume of ethyl acetate? On the contrary it would be understood that the 15 kg were processed with an equal volume of ethyl acetate obtaining a 500 g extract. Please explain.

Response: Thank you for pointing this out. We are deeply sorry for this misunderstanding caused by our improper expression. Based on your opinion, we have modified the original content as follows: Under sterile conditions, the fungus Xylaria sp. VDL4 cultured in PSA medium was inoculated into sterile 500 mL rice fermentation bottles. Each bottle contained 70.0 g rice and 70.0 mL water. A total of 15.0 kg rice was apportioned into about 200 containers and subsequently inoculated. The inoculated samples were then cultured in a constant temperature incubator at 25℃ for 30 days. Following fermentation, the mixture was carefully decanted from the fermentation bottles and transferred to a barrel. Subsequently, the mixture underwent extraction through crushing and soaking in ethyl acetate for 24 hours at room temperature, The extraction process was repeated six times, and the resulting filtrates were combined. The combined filtrate was dried and concentrated to obtain approximately the extract (500 g). This modification can be found on line 111 of Section 2.3, Page 3 in the revised manuscript. 

  1. In section 2.6.2 In Vivo Antifungal Activity Assay… The authors describe the methodology for the bacterium Staphylococcus aureusnot for the pathogenic fungi ( oxysporum, B. cinerea, P. capsici, and F. solani).

Response: Thank you for pointing this out. We deeply apologize for wasting your time. Our carelessness led to incorrect data in section 2.6.2. In this section, titled "In Vivo Antifungal Activity Assay", what we intended to describe was the method related to Botrytis cinerea, not Staphylococcus aureus. We have made the necessary revisions in the manuscript. This modification can be found on line 204 of Section 2.6.2, page 5 of the revised manuscript.

  1. In line 244 the authors mentioned… Specially, its inhibitions were superior to the positive 244 control carbendazim (MIC = 15.62 μg/ml) for oxysporum and thiabendazole (MIC = 25.62 245 μg/ml) for P. capsici… But the thiabendazole for P. capsici in Table 2 is 15.62 μg/ml not 25.62 μg/ml.

Response: We appreciate your comments on the manuscript. We deeply apologize for wasting your time. Our carelessness led to the wrong data mentioned by the authors in line 244. We have made the following modifications in the manuscript. Specifically, for F. oxysporum, its inhibitions were superior to those of the positive control carbendazim (MIC = 15.62 μg/ml), and for P. capsici, they were superior to those of the positive control thiabendazole (MIC = 15.62 μg/ml). This modification can be found on line 316 of Section 3.2, page 9 of the revised manuscript.

  1. Correct the name of the fungi Phytophthora capsicain table 2.

Response: Thank you again for pointing this out. In the manuscript, we have corrected the misnamed Phytophthora capsica to Phytophthora capsici. This modification is located in Table 2 of Section 3.2 on Page 10 of the manuscript.

Reviewer 2 Report

The manuscript communicates a relevant issue, the search for natural compounds with antifungal properties. even though the idea to be presented is understandable, the manuscript contains several aspects that need improvement:

- It is not demonstrated whether the compounds have fungistatic or fungicide activity, this needs additional experimentation.

- Because of the potential use in farming fields, the cytotoxicity of compounds against human, mammalian, and plant cells should be addressed.

- The methodology needs a major improvement. Currently, most of the methodologies lack details allowing experiment replication; in particular the analytical methods.

- It is not disclosed the source of glucosidase, please amend.

-As control of the glucosidase activity assays, an irrelevant enzyme should be included.

-There is no rationale for the analysis of the effect of these compounds on glucosidase activity, please clarify. Why was a beta-glucosidase activity not tested?

-The manuscript lacks a proper statistical analysis of data, a proper description of results, and a discussion.

The manuscript communicates a relevant issue, the search for natural compounds with antifungal properties. even though the idea to be presented is understandable, the manuscript contains several aspects that need improvement:

- It is not demonstrated whether the compounds have fungistatic or fungicide activity, this needs additional experimentation.

- Because of the potential use in farming fields, the cytotoxicity of compounds against human, mammalian, and plant cells should be addressed.

- The methodology needs a major improvement. Currently, most of the methodologies lack details allowing experiment replication; in particular the analytical methods.

- It is not disclosed the source of glucosidase, please amend.

-As control of the glucosidase activity assays, an irrelevant enzyme should be included.

-There is no rationale for the analysis of the effect of these compounds on glucosidase activity, please clarify. Why was a beta-glucosidase activity not tested?

-The manuscript lacks a proper statistical analysis of data, a proper description of results, and a discussion.

Author Response

To reviewer #2:

Major comments and Detail comments:

The manuscript communicates a relevant issue, the search for natural compounds with antifungal properties. even though the idea to be presented is understandable, the manuscript contains several aspects that need improvement:

  1. It is not demonstrated whether the compounds have fungistatic or fungicide activity, this needs additional experimentation.

Response: First of all, we sincerely appreciate your professional advice. Based on the hypothesis that plant endophytic fungi can protect host plants from pathogen invasion[1]. this study investigates the antifungal properties of secondary metabolites produced by Xylaria sp. isolated from Vaccinium dunalianum. In this study, we identified a series of secondary metabolites that represent novel compounds not previously documented in the literature. To date, there has been no investigation into their potential antifungal properties. Consequently, we undertook a systematic evaluation of their antifungal activities. The antifungal experiments revealed that the newly identified compounds 1, 2, and 3 exhibited varying degrees of inhibitory activity against plant pathogenic fungi (Fusarium oxysporum, Botrytis cinerea, Phytophthora capsici, and Fusarium solani). As shown in table 2, compound 1-3 showed significant inhibitions against four plant pathogenic fungi with  minimum inhibitory concentration (MIC) values ranged from 3.91 to 31.25 μg/mL (Table 2). Among them, xylariside B (2) had the strongest inhibitory activity against the tested pathogenic fungi. Notably, for Fusarium oxysporum, its inhibitory effect was superior to that of the positive control carbendazim (MIC=15.62 μg/mL); for Phytophthora capsici, its inhibitory effect was superior to that of the positive control thiabendazole (MIC=15.62 μg/mL). This preliminarily demonstrates that the compound has antifungal activities in vitro. The plant-pathogenic fungus Botrytis cinerea is widely recognized as a highly harmful pathogen that can cause gray mold on a wide variety of vegetables, fruits, and ornamental plants [2, 3]. To prevent the occurrence of gray mold in tomatoes and strawberries, based on the antifungal performance observed in laboratory tests, we further evaluated the effectiveness of xylariside A-C (1-3) against Botrytis cinerea in vivo. As shown in Figures 4 and 5, compound 2 exhibited a significant protective effect at a concentration of 250.0 μg/mL. Compared with the blank control group, the lesion area in the protected group was significantly reduced, and the effect was equivalent to that of the positive control. This further confirms that the compounds reported in the manuscript have fungistatic activities.

  1. Wang, W.-H.; Li, C.-R.; Qin, X.-J.; Yang, X.-Q.; Xie, S.-D.; Jiang, Q.; Zou, L.-H.; Zhang, Y.-J.; Zhu, G.-L.; Zhao, P. Novel Alkaloids from Aspergillus Fumigatus VDL36, an Endophytic Fungus Associated with Vaccinium Dunalianum. Agric. Food Chem. 2024, 72, 10970–10980, doi: 10.1021/acs.jafc.4c00371.
  2. Chen, Y.-J.; Liu, H.; Zhang, S.-Y.; Li, H.; Ma, K.-Y.; Liu, Y.-Q.; Yin, X.-D.; Zhou, R.; Yan, Y.-F.; Wang, R.-X.; et al. Design, Synthesis, and Antifungal Evaluation of Cryptolepine Derivatives against Phytopathogenic Fungi. J. Agric. Food Chem. 2021, 69, 1259–1271, doi:10.1021/acs.jafc.0c06480.
  3. Jiao, W.; Liu, X.; Chen, Q.; Du, Y.; Li, Y.; Yue, F.; Dong, X.; Fu, M. Epsilon-Poly-l-Lysine (ε-PL) Exhibits Antifungal Activity in Vivo and in Vitro against Botrytis Cinerea and Mechanism Involved. Postharvest Biology and Technology 2020, 168, 111270, doi:10.1016/j.postharvbio.2020.111270.
  4. Because of the potential use in farming fields, the cytotoxicity of compounds against human, mammalian, and plant cells should be addressed

Response: Thank you for your professional suggestions. In fact, the work mentioned in your comments is exactly what we plan to carry out in the subsequent stages. However, due to the current technical limitations, these tasks have not been actually implemented yet. We sincerely appreciate your understanding and tolerance.

  1. The methodology needs a major improvement. Currently, most of the methodologies lack details allowing experiment replication; in particular the analytical methods.

Response: Thank you very much for pointing this out. We fully agree with your comments. In view of the questions you raised, we have revised the methods described in Sections 2.2, 2.3, 2.6.1, 2.6.2, and 2.6.3 of the manuscript. In the revised manuscript, the specific locations are as follows: Page 2, Line 80; Page 3, Line 106; Page 4, Line 162; Page 5, Line 197; and Page 6, Lines 216 and 230.

  1. It is not disclosed the source of glucosidase, please amend.
  2. Response: Thank you for pointing this out. We agree with this comment. The α-glucosidase employed in the study was sourced from Saccharomyces cerevisiae. We have incorporated this information regarding the origin of the α-glucosidase into the manuscript. In the revised version, this modification can be located at line 230 of Section 2.6.3, page 6.
  3. As control of the glucosidase activity assays, an irrelevant enzyme should be included. There is no rationale for the analysis of the effect of these compounds on glucosidase activity, please clarify. Why was a beta-glucosidase activity not tested?

Response: Thank you for pointing this out. We extend our sincere gratitude for your insightful and professional suggestions. During the execution of the α-glucosidase activity inhibition assay, both the α-glucosidase enzyme and the substrate, p-nitrophenyl-α-D-glucopyranoside (PNPG, batch number S10137), were procured from Shanghai Yuanye Biotechnology Co., Ltd. This enzyme-substrate pairing was meticulously selected based on well-established biochemical principles, ensuring a high degree of compatibility. Prior to commencing the full-scale experiment, a series of preliminary trials were conducted. In one set of pre-experiments, we reacted p-nitrophenyl-β-D-glucopyranoside (batch number S10137) with α-glucosidase. As expected from the known substrate-specificity of α-glucosidase, no enzymatic activity was detected. Conversely, when α-glucosidase was reacted with p-nitrophenyl-α-D-glucopyranoside (batch number S10137), distinct color changes were observed in the enzyme solutions at concentrations of 0.2 U/mL and 1 U/mL. These colorimetric changes are direct indicators of the hydrolytic activity of α-glucosidase on the substrate. The observed enzyme activity at these concentrations further validates the proper functioning of the enzyme and the suitability of the substrate. To further confirm the specificity of the observed enzymatic reactions, we also carried out control experiments where amylase was reacted with p-nitrophenyl-α-D-glucopyranoside (batch number S10137). As anticipated, no enzymatic activity was produced, which aligns with the well-characterized substrate-specificity profiles of amylase. Consequently, it is scientifically justifiable to conduct an in-depth analysis of the impacts exerted by these compounds on glucosidase activity.

  1. The manuscript lacks a proper statistical analysis of data, a proper description of results, and a discussion.

Response: Thank you for pointing this out. We agree with this comment. Therefore, to make the article more complete and easier for readers to understand, we have added an analysis of the experimental results at the corresponding positions in the manuscript and supplemented the content of the discussion section. If you check the revised manuscript, this modification appears on line 342 of Section 3.4, page 12, and line 355 of Section 4, page 13.

Reviewer 3 Report

The manuscript presents important findings for the area and with prospects for practical application. The authors were succinct and limited in the description of the results, providing little discussion about similar studies in the area and with similar compounds.

Some points that authors should take into account before publication:

1- Describe the role of alpha-glucosidase in the introduction of the text.

2- Correct the word "argiculture" on line 75.

3- In the methodology, describe the concentration range of the substances used to determine the MIC, both in in vivo and in vitro studies (items 2.6.1 and 2.6.2).

4- S. aureus is only mentioned in the methodology (2.6.2). Was there a subsequent test to assess the microbial load in the fruit infection?

5- The titles of the tables and figures should be complete, providing enough information to understand them, such as the full name of the compounds and not just numbers (1, 2 and 3). Information on the meanings of acronyms should be defined at the table footnote. Tables should be as comprehensible as possible on its own, without the need to refer to the text to understand details such as codes and acronyms.

6- The authors fail to discuss the text. I recommend checking similar studies in the literature with similar compounds, to enrich the argument of the study's practical applicability.

Author Response

To reviewer #3:

Major comments:

The manuscript presents important findings for the area and with prospects for practical application. The authors were succinct and limited in the description of the results, providing little discussion about similar studies in the area and with similar compounds.

Response: Thank you for pointing this out. We agree with this comment. To make the article more complete and easier for readers to understand, we have added content to the discussion section in the manuscript and further analyzed the experimental results. If you look at the revised manuscript, this modification appears on line 355 of Section 4, Page 13.

Detail comments:

Some points that authors should take into account before publication:

  1. Describe the role of alpha-glucosidase in the introduction of the text.

Response: Thank you for pointing this out. We agree with this comment. To make the article more complete, we have added the following content at section 3.4 of the article according to your suggestions: Diabetes Mellitus, a metabolic disorder characterized by elevated blood glucose levels, has emerged as a major global health concern, precipitating long-term complications and incurring substantial economic burdens. α-Glucosidase is involved in the hydrolysis of starch into monosaccharides and disaccharides, a process that elevates blood glucose levels. Thus, the inhibition of α-glucosidase is pivotal in diabetes treatment. Prior literature indicates that polyacetylene compounds can inhibit α-glucosidase. This demonstrates the role of α-glucosidase. The present study was gradually advanced based on the theory of the protection of host plants by plant endophytic fungi. Given that the core of the research focused on the protective relationship between plant endophytic fungi and host plants, in the "Introduction" section, we mainly elaborated on this core theme, and thus did not discuss the inhibitory activity of α-glucosidase. It is worth noting that the inhibitory activity of these new compounds against glucosidase had not been evaluated by relevant research before. Considering the need to understand the functional characteristics of these new molecules more comprehensively from multiple dimensions, we carried out the evaluation of this activity. To further improve the content system of the article, making it more coherent logically and more complete content-wise, we added relevant elaborations on the role of glucosidase in section 3.4, hoping to provide readers with more comprehensive information. Certainly, in the revised manuscript, you can find this modification at line 342 of Section 3.4 on page 12.

  1. Correct the word "argiculture" on line 75.

Response: Thank you for pointing this out. I am sorry for such an error, and you pointed out that the word "argiculture" on line 75 should be "agriculture". I have corrected accordingly. you can find this modification at line 99 of Section 2.2 on page 3.

  1. In the methodology, describe the concentration range of the substances used to determine the MIC, both in in vivoand in vitro studies (items 2.6.1 and 2.6.2).

Response: Thank you for pointing this out. We agree with this comment. Therefore, in the methodology section, we have provided a detailed description of the concentration range of the substances used to determine the Minimum Inhibitory Concentration (MIC). The in vitro study methods primarily involve antifungal activity assessment using the microdilution broth method. The specific modifications are as follows:The antifungal activities of compounds (1-3) were evaluated in vitro using the microbroth dilution method in 96-well plates with PDA medium. Thiabendazole and carbendazim, provided by Aladdin Chemical Co., Ltd., were used as positive controls, while an equivalent concentration of dimethyl sulfoxide (DMSO) solution served as the negative control. The fungi were cultured in PD broth at 28 ± 0.5°C for 48 hours, and then spore suspensions were prepared by dilution in PDB broth to a concentration of approximately 1 × 10^6 CFU/mL. The dilution experiments were conducted using 96-well plates. In the first column of each row, 50 μL of stock solution (mother liquor) and 100 μL of PDA medium were added to ensure the stock solution remained uncontaminated. In the second column, 100 μL of stock solution was added. From the third to the twelfth columns, 50 μL of medium was added. Then, 50 μL of the stock solution from the second column was transferred to the third column, thoroughly mixed, and sequentially transferred to the fourth column, continuing this process until the eleventh column. The 50 μL removed from the eleventh column was discarded into a waste container. Finally, 100 μL of fungal suspension was added to each well from the second to the twelfth columns, with the twelfth column serving as a control to confirm that the fungal suspension was uncontaminated. The compound concentrations started at an initial value of 250 μg/mL and were serially diluted nine times, reaching a minimum concentration of 0.49 μg/mL. Each compound was tested in duplicate. After incubation at 28 ± 0.5°C for 48 hours, the growth of pathogenic fungi was observed(Figure S37). The minimum inhibitory concentration (MIC) was defined as the lowest concentration of the test compound in each well at which no microbial growth was observed. In the in vivo experiments, each compound was used at a concentration of 250 μg/mL. Since we did not measure the MIC in the fruit model, we have confirmed that these compounds exhibit good inhibitory effects on the tested pathogens. In the revised manuscript, you can find this modification on line 162 of Section 2.6.1, page 5, and on line 207 of Section 2.6.2, page 5.

  1. aureus is only mentioned in the methodology (2.6.2). Was there a subsequent test to assess the microbial load in the fruit infection?

Response: Thank you for pointing this out. We deeply apologize for wasting your time. Our carelessness led to incorrect data in section 2.6.2. In this section, titled "In Vivo Antifungal Activity Assay", what we intended to describe was the method related to Botrytis cinerea, not Staphylococcus aureus. We have made the necessary revisions in the manuscript. In the revised manuscript, you can find this modification at line 204 of Section 2.6.2 on page 5.

  1. The titles of the tables and figures should be complete, providing enough information to understand them, such as the full name of the compounds and not just numbers (1, 2 and 3). Information on the meanings of acronyms should be defined at the table footnote. Tables should be as comprehensible as possible on its own, without the need to refer to the text to understand details such as codes and acronyms.

Response: Thank you for pointing this out. We agree with this comment. Therefore, we have provided explanations for the meanings of the abbreviations in the footnotes of the table. Meanwhile, we have optimized the titles of the tables and figures. You can find this modification on line 322 of Section 3.2, page 10, on line 339 of Section 3.3, page 12 and on line 354 of Section 3.4, page 13.

  1. The authors fail to discuss the text. I recommend checking similar studies in the literature with similar compounds, to enrich the argument of the study's practical applicability.

Response: Thank you for pointing this out. We agree with this comment. Therefore, to ensure the completeness of the article, we have added a discussion section to the manuscript. In the revised manuscript, this modification is located on line 355 of Section 4 page 13.

Reviewer 4 Report

This study isolated three new quinovopyranoside derivatives from fungus Vaccinium dunalianum Wight and evaluated their antifungal and anti-glucosidase properties. I consider that it describes complete experimental research, and the document is well prepared. The manuscript can be accepted for publication, with minor observations.

·       At line 51, Write the names of phytopathogenic fungi without point (Fusarium. oxysporum, Botrytis. Cinerea, Phytophthora, capsica, and Fusarium. solani).

·       At line 122, in Activity Determination of Compounds (1-3) section.

·       The word “in vitro”, write in italic.

·       Please, add the tested concentration range of each compound for the in vitro and in vivo assays.

·       In the In vivo antifungal activity assay section, confirm if the microorganism Staphylococcus aureus was evaluated, since in Results section, the fungistatic effects of the new compounds on the B. cinerea fungus is discussed.

·       At line 165, mention the source of origin of the α-glucosidase (Saccharomyces cerevisiae, Bacillus, stearothermophilus).

·       In Results section, please homogenize the units “µg/mL” and “IC50” in their manuscript.

·       In Figure 5, add the concentration assessed for positive controls, and confirm whether it is a bacteriostatic effect or a fungistatic effect.

·       At the end of the Results section, it would be interesting to mention the relevance of the conformational change of the quinovopyranoside moiety in compound 2, since its activity increases significantly in all assays compared to 1.

The image quality of Figures 4 and 5 could be improved.

Author Response

Major comments:

This study isolated three new quinovopyranoside derivatives from fungus Vaccinium dunalianum Wight and evaluated their antifungal and anti-glucosidase properties. I consider that it describes complete experimental research, and the document is well prepared. The manuscript can be accepted for publication, with minor observations.

  1. At line 51, Write the names of phytopathogenic fungi without point (Fusarium. oxysporum, Botrytis. Cinerea, Phytophthora, capsica, and Fusarium. solani).

Response: Thank you for pointing this out. We agree with this comment. Therefore, we have corrected the names of the four plant pathogenic fungi in the manuscript to Fusarium oxysporum, Botrytis cinerea, Phytophthora capsici, and Fusarium solani. In the revised manuscript, this modification is located on line 61 of Section 1 page 2.

  1. At line 122, in Activity Determination of Compounds (1-3) section. The word “in vitro”, write in italic.

Response: Thank you for pointing this out. We agree with this comment. Therefore, we deeply regret the oversight that occurred, and we have already made the necessary amendments at the corresponding positions within the manuscript. In the revised manuscript, this modification is located on line 197 of Section 2.6.2 page 5.

  1. Please, add the tested concentration range of each compound for the in vitro and in vivo assays.

Response: Thank you for pointing this out. We agree with this comment. As per your request, we have added the test concentration range for each compound in both the in vitro and in vivo experiments in the methodology section: In vitro experiments: The concentration range for each compound tested was from 250 μg/mL (initial concentration) to 0.49 μg/mL, after nine successive dilutions. In vivo experiments: In in vivo experiments, the concentration of each compound used was 250 μg/mL, as we did not measure the minimum inhibitory concentration (MIC) in the fruit model. However, we have confirmed that these compounds exhibit good inhibitory effects on the tested pathogens. In the revised manuscript, you can find this modification on line 162 of Section 2.6.1, page 5, and on line 207 of Section 2.6.2, page 5.

  1. In the In vivoantifungal activity assay section, confirm if the microorganism Staphylococcus aureus was evaluated, since in Results section, the fungistatic effects of the new compounds on the cinerea fungus is discussed.

Response: Thank you for pointing this out. We deeply apologize for wasting your time. Our carelessness led to incorrect data in section 2.6.2. In this section, titled "In Vivo Antifungal Activity Assay", what we intended to describe was the method related to Botrytis cinerea, not Staphylococcus aureus. We have made the necessary revisions in the manuscript. In the revised manuscript, you can find this modification at line 204 of Section 2.6.2 on page 5.

  1. At line 165, mention the source of origin of the α-glucosidase (Saccharomyces cerevisiae, Bacillus, stearothermophilus).

Response: Thank you for pointing this out. We agree with this comment. Therefore, in the manuscript, we added an explanation that α-glucosidase is derived from Saccharomyces cerevisiae. you can find this modification at line 230 of Section 2.6.3 on page 6.

  1. In Results section, please homogenize the units “µg/mL” and “IC50” in their manuscript.

Response: Thank you for pointing this out. We agree with this comment. As per your request, Therefore, we have made modifications at the corresponding positions in the manuscript. You can find this modification in Table 3 of Section 3.4 on page 13.

  1. In Figure 5, add the concentration assessed for positive controls, and confirm whether it is a bacteriostatic effect or a fungistatic effect.

Response: Thank you for pointing this out. We agree with this comment. Therefore, the concentration of positive controls has been incorporated into the relevant sections of the manuscript, and it has been ascertained that the observed effect is fungistatic. You can find this modification at line 339 of Section 3.3 on page 12.

  1. At the end of the Results section, it would be interesting to mention the relevance of the conformational change of the quinovopyranoside moiety in compound 2, since its activity increases significantly in all assays compared to 1.

Response: Thank you for pointing this out. We agree with this comment. Therefore, we added a discussion section to the manuscript and mentioned this content therein. You can find this modification at line 372 of Section 4 on page 13.

Detail comments:

  1. The image quality of Figures 4 and 5 could be improved.

Response: Thank you for pointing this out. We agree with this comment. Therefore, we have improved the image quality of Figures 4 and 5. You can find these two modifications at line 333 of Section 3.3 on page 11 and at line 336 of Section 3.3 on page 12.

Reviewer 5 Report

The article «Antifungal Polyacetylenic Deoxyglycosides Isolated from Endophytic Fungus Xylaria sp. VDL4 Associated with Vaccinium dunalianum» is an interesting experimental work aimed at identifying natural compounds with antifungal activity. The authors identified three compounds from the fungus Xylaria sp. VDL4, characterized their composition and chemical formula, and tested their antifungal activity. It was shown that these compounds may be promising for further use.

I believe that the article can be published in the Journal of Fungi after revision.

Below are my comments and suggestions for improving the text.

Abstract. The abbreviation MIC should be deciphered here.

Introduction.

In the introduction, the authors provide very little information about natural polyacetylenes. They provide several references, but do not disclose the current state of affairs. Most of the introduction is devoted to the purpose and description of the authors' work.

I think it is worth adding information about:

Targets of polyacetylenes in fungal cells. Their activity against other plant pathogens. Toxicity to animals and humans.

It is necessary to justify why the fungus Xylaria 44 sp. VDL4, isolated from the leaves of Vaccinium dunalianum, was chosen as the object of the study.

 Methodology.

point 2.2 «This species was determined through the analysis of both morphological characteristics and ITS sequences, exhibiting a similarity of 99%.» Lines 71-72. What species (fungus or plant) are we talking about? Specify.

point 2.3

Please describe the cultivation of the fungus in more detail and in greater detail. What kind of medium with 15 kg of rice is this?

How was the «total extract (500 g)» obtained?

point 2.6.1 Lines 130-134 Check the description of the preparation of fungal spores and their treatment with the analyzed compounds. How were the spores further cultivated? It is not clear now. By what parameter was the MIC determined?

point 2.6.3.

«Different concentrations of the purified compounds (20.0 μL) were mixed with the enzyme solution (50.0 μL) and incubated at 37 °C for 15 minutes.» Lines 151-152. The concentrations used must be given.

Lines 166-170. Please check these sentences. What does «These reagents, intended for reducing blood glucose levels, were sourced from Shanghai Yuanye Biotechnology Co., Ltd. Additionally» mean? «using a UV-5900 enzyme-labeled instrument»?

Results

point 3.2 Results of the In Vitro Antifungal Activity Assay. What parameter was used to estimate the MIC? It is worth providing images of germinating spores or fungal colonies. Statistics?

point 3.3. Provide statistics. How many tomatoes and strawberries were processed? In how many replicates?

 point 3.4. Nowhere in the article is the choice of the enzyme α-glucosidase justified. Why is this enzyme being studied?

Discussion

The article lacks a discussion section. It is necessary to have one separately or the discussion should be present in the results section. It is necessary to compare the obtained results with those already known.

Author Response

To reviewer #5:

Major comments:

The article «Antifungal Polyacetylenic Deoxyglycosides Isolated from Endophytic Fungus Xylaria sp. VDL4 Associated with Vaccinium dunalianum» is an interesting experimental work aimed at identifying natural compounds with antifungal activity. The authors identified three compounds from the fungus Xylaria sp. VDL4, characterized their composition and chemical formula, and tested their antifungal activity. It was shown that these compounds may be promising for further use. I believe that the article can be published in the Journal of Fungi after revision.

Detail comments: Below are my comments and suggestions for improving the text.

  1. The abbreviation MIC should be deciphered here.

Response: Thank you for pointing this out. We agree with this comment. Therefore, to make it easier for readers to understand, we have added the specific meaning of the abbreviation "MIC" wherever it appears in the abstract. You can find this modification at line 22 of the Abstract section on page 1.

  1.  In the introduction, the authors provide very little information about natural polyacetylenes. They provide several references, but do not disclose the current state of affairs. Most of the introduction is devoted to the purpose and description of the authors' work. I think it is worth adding information about: Targets of polyacetylenes in fungal cells. Their activity against other plant pathogens. Toxicity to animals and humans. It is necessary to justify why the fungus Xylariasp. VDL4, isolated from the leaves of Vaccinium dunalianum, was chosen as the object of the study.

Response: Thank you for pointing this out. We agree with this comment. Therefore, we have revised the introduction part according to your opinion. You can find this modification at line 37 of the Introduction section on page 1.

  1.  point 2.2 «This species was determined through the analysis of both morphological characteristics and ITS sequences, exhibiting a similarity of 99%.» Lines 71-72. What species (fungus or plant) are we talking about? Specify.

Response: Thank you for pointing this out. We agree with this comment. According to these suggestions, we have supplemented the separation methods, morphological features, and molecular characterization of Xylaria VDL4 at the corresponding positions in the manuscript. Modify as follows: The endophyte Xylaria sp. VDL4 was isolated from the healthy leaf tissues of Vaccinium dunalianum collected in Wuding County, Yunnan Province, China in 2016. After collection, the leaves were strictly disinfected to remove contaminants. The disinfected leaves were cut into 0.5cm×0.5cm pieces and inoculated on Potato-Sucrose Agar (PSA) medium. The inoculated medium was incubated at 28 °C in  dark for about 7 days. When mycelium emerged on the leaf-piece edges, the mycelium's front end was selected and transferred to fresh PSA medium for further growth. After 3-4 generations of purification culture, a pure strain was obtained. The strain's colony had an uneven, convex - centered appearance, with white mycelium and oval spores. To identify the strain, DNA of VDL4 was extracted and amplified via PCR using fungal universal primers. The PCR products were analyzed by agarose gel electrophoresis, sequenced, and the results were used for a homology search in the NCBI database (blaster.ncbi.nlm.nih.gov). The search showed 100% homology with Xylaria sp., thus identifying the bacterium as Xylaria sp (Figure33). Therefore, the species we refer to is Xylaria VDL4. You can find this modification at line 80 of Section 2.2 on page 2.

  1. point 2.3 Please describe the cultivation of the fungus in more detail and in greater detail. What kind of medium with 15 kg of rice is this? How was the «total extract (500 g)» obtained?

Response: Thank you for pointing this out. We are deeply sorry for this misunderstanding caused by our improper expression. Based on your opinion, we have modified the original content as follows: Under sterile conditions, the fungus Xylaria sp. VDL4 cultured in PSA medium was inoculated into sterile 500 mL rice fermentation bottles. Each bottle contained 70.0 g rice and 70.0 mL water. A total of 15.0 kg rice was apportioned into about 200 containers and subsequently inoculated. The inoculated samples were then cultured in a constant temperature incubator at 25℃ for 30 days. Following fermentation, the mixture was carefully decanted from the fermentation bottles and transferred to a barrel. Subsequently, the mixture underwent extraction through crushing and soaking in ethyl acetate for 24 hours at room temperature, The extraction process was repeated six times, and the resulting filtrates were combined. The combined filtrate was dried and concentrated to obtain approximately the extract (500 g). This modification can be found on line 111 of Section 2.3, Page 3 in the revised manuscript.

  1. point 2.6.1 Lines 130-134 Check the description of the preparation of fungal spores and their treatment with the analyzed compounds. How were the spores further cultivated? It is not clear now. By what parameter was the MIC determined?

Response: Thank you for pointing this out. We agree with this comment. Based on your feedback, we have provided more detailed explanations on the preparation of fungal spores, handling of compounds, and subsequent culture process, as well as clarified the parameters for MIC determination. The details are as follows:

Preparation of fungal spores and compound handling: The fungi used in this study were cultured in PD broth at 28 ± 0.5 °C for 24 hours. The spore suspension was diluted with PD broth to a concentration of approximately 1 × 10^6 CFU/mL. The test compounds were prepared in DMSO at a concentration of 0.25 mg/mL. Commercial fungicides ketoconazole and carbendazim were used as positive controls, while an equivalent concentration of DMSO solution served as the negative control.

Spore culture and MIC determination: Using a 96-well plate, 50 μL of stock solution and 100 μL of PDA medium were added to the first column of each row to ensure the stock solution was uncontaminated. In the second column, 100 μL of stock solution was added. From the third to the twelfth columns, 50 μL of medium was added. Subsequently, 50 μL of stock solution from the second column was transferred to the third column, mixed thoroughly, and then sequentially transferred to the fourth column, continuing this process up to the eleventh column. The 50 μL removed from the eleventh column was discarded into a waste container. Finally, 100 μL of fungal suspension was added to each well from the second to the twelfth columns. The compound concentration ranged from 250 μg/mL (initial concentration) and was serially diluted nine times to reach a minimum concentration of 0.49 μg/mL. Two sets of replicate experiments were performed for each compound.

Commercial fungicides ketoconazole and carbendazim were used as positive controls, while an equivalent concentration of DMSO solution served as the negative control. After incubation at 28 ± 0.5 °C for 48 hours, the growth of pathogens was observed, and the MIC was determined as the lowest concentration of the test compound at which no microbial growth was observed.

We hope these additions and clarifications address your concerns. Thank you again for your review and suggestions. You can find this modification at line 162 of Section 2.6.1 on page 5.

       6. point 2.6.3. «Different concentrations of the purified compounds (20.0μL) were mixed with the enzyme solution (50.0 μL) and incubated at 37 °C for 15 minutes.» Lines 151-152. The concentrations used must be given.

Response: Thank you for pointing this out. We agree with this comment. Since different compounds have varying inhibitory effects on α-glucosidase, we set different concentration gradients for each compound during the experiment. This required us to determine the appropriate concentration gradient for each compound through experimentation. Following your suggestion, we have added the concentration ranges and inhibition rates of acarbose and Compounds 1-3 in the supporting materials (Figure S36). You can find this modification in Figure 36 of the supporting materials.

  1. Lines 166-170. Please check these sentences. What does «These reagents, intended for reducing blood glucose levels, were sourced from Shanghai Yuanye Biotechnology Co., Ltd. Additionally» mean? «using a UV-5900 enzyme-labeled instrument»?

Response: Thank you for pointing this out. We agree with this comment. Therefore, we have made revisions at the corresponding positions in the manuscript regarding the issues you raised. The details are as follows: The water used in the experiment was ultrapure water. α-glucosidase which is extracted from Saccharomyces cerevisiae and Bacillus stearothermophilus, PBS phosphate buffer, p-nitrophenyl-α-D-glucopyranoside (batch number S10137), and acarbose hydrate (98%) were purchased from Shanghai Yuanye Biotechnology Co., Ltd. DMSO was obtained from Beijing Solarbio Science & Technology Co., Ltd. The SpectraMax 190 microplate reader used in this experiment was produced by Shanghai Molecular Devices Co., Ltd. You can find this modification at line 230 of Section 2.6.3 on page 6.

  1.  Results point 3.2 Results of the In Vitro Antifungal Activity Assay. What parameter was used to estimate the MIC? It is worth providing images of germinating spores or fungal colonies. Statistics?

Response: Thank you for pointing this out. We agree with this comment. Therefore, we have made modifications to the content you mentioned in the manuscript, as follows: The fungi used in this study were cultured on PD medium at 28 ± 0.5°C for 24 hours. The spore suspension was then diluted with PD medium to approximately 1 × 10^6 CFU/mL. The test compounds were dissolved in DMSO at a concentration of 0.25 mg/mL. Commercial fungicides, ketoconazole and benomyl, were used as positive controls, while DMSO solution at the same concentration served as the negative control. Using a 96-well plate, 50 μL of the stock solution and 100 μL of PDA medium were added to the first column of each row as the control group to ensure that the stock solution was not contaminated. In the second column, 100 μL of the stock solution was added. From the third to the twelfth columns, 50 μL of medium was added to each well. Subsequently, 50 μL of the stock solution from the second column was transferred to the third column, mixed thoroughly, and then 50 μL was sequentially transferred to the fourth column, continuing in this manner until the eleventh column. The final 50 μL in the eleventh column was discarded. Finally, 100 μL of fungal suspension was added to each well from the second to the twelfth columns. The compound concentrations ranged from 250 μg/mL (initial concentration) with nine consecutive dilutions, reaching a final concentration of 0.49 μg/mL. Each compound was tested in duplicate experiments. Commercial fungicides, ketoconazole and benomyl, served as positive controls, and DMSO solution at the same concentration served as the negative control. After incubation at 28 ± 0.5°C for 48 hours, the growth of pathogens was assessed visually, and the minimum inhibitory concentration (MIC) was determined as the lowest concentration at which no microbial growth was observed. Regarding the images of spore germination or fungal colonies, in order to further illustrate the experimental results, we have included the revised support materials (Figure S37) in the list. Moreover, we will conduct statistical analyses of the experimental results to improve the reliability and reproducibility of the data. The above - mentioned modifications can be found at line 162 of Section 2.6.1 on page 5 and in Figure 37 of the supporting materials.

  1. point 3.3. Provide statistics. How many tomatoes and strawberries were processed? In how many replicates?

Response: Thank you for pointing this out. We agree with this comment. Therefore, to facilitate readers' comprehension, we have supplemented the following content in Section 3.3 of the manuscript. In this experiment, a total of 54 tomato and strawberry fruits of uniform size were care-fully selected. These fruits were then randomly allocated into six distinct groups. Each of these groups encompassed nine fruits and was subsequently partitioned into three rep-licate subgroups, with each subgroup holding three fruits. You can find this modification at line 197 of Section 2.6.2 on page 5.

  1. point 3.4. Nowhere in the article is the choice of the enzyme α-glucosidase justified. Why is this enzyme being studied?

Response: Thank you for pointing this out. We agree with this comment. Therefore, to make the article more complete, we have added the following content at section 3.4 of the article according to your suggestions: Diabetes Mellitus, a metabolic disorder characterized by elevated blood glucose levels, has emerged as a major global health concern, precipitating long-term complications and incurring substantial economic burdens. α-Glucosidase is involved in the hydrolysis of starch into monosaccharides and disaccharides, a process that elevates blood glucose levels. Thus, the inhibition of α-glucosidase is pivotal in diabetes treatment. Prior literature indicates that polyacetylene compounds can inhibit α-glucosidase. You can find this modification at line 342 of Section 3.4 on page 12.

  1.  The article lacks a discussion section. It is necessary to have one separately or the discussion should be present in the results section. It is necessary to compare the obtained results with those already known.

Response: Thank you for pointing this out. We agree with this comment. Therefore, to ensure the completeness of the article, we have added a discussion section to the manuscript. You can find this modification at line 355 of Section 4 on page 13.

Round 2

Reviewer 1 Report

The authors satisfactory improve their paper. So can be accepted in the present form.

No further comments.

Author Response

1. The authors satisfactory improve their paper. So can be accepted in the present form.

We sincerely appreciate your positive evaluation and kind approval of our paper. Your recognition of our efforts in improving the manuscript means a great deal to us. It has been a challenging yet rewarding journey to refine the paper to its current state, and your feedback has been instrumental in this process.

If there is any further administrative procedure or minor adjustment you'd like us to attend to, please do not hesitate to let us know. We are committed to ensuring that the final publication of our work meets all the requirements and standards of the journal.

Once again, thank you for your time, expertise, and valuable input.

Reviewer 2 Report

The authors improved the manuscript but still, some points need attention:

  • The authors agreed that establishing compounds' fungistatic/fungicide effect is essential. So, there is no scientific justification for leaving this aspect out of the manuscript.
  • The statistical analysis needs to be incorporated into the results. A section indicating the methodology to establish statistical significance is required.

The authors improved the manuscript but still, some points need attention:

  • The authors agreed that establishing compounds' fungistatic/fungicide effect is essential. So, there is no scientific justification for leaving this aspect out of the manuscript.
  • The statistical analysis needs to be incorporated into the results. A section indicating the methodology to establish statistical significance is required.

Author Response

1. The authors agreed that establishing compounds' fungistatic/fungicide effect is essential. So, there is no scientific justification for leaving this aspect out of the manuscript.

Response: We are extremely grateful to the reviewers for their valuable comments. We fully agree that the evaluation of fungistatic/fungicide effects is crucial for a comprehensive understanding of the antifungal potential of the compounds. However, due to limitations in experimental conditions and time, this study mainly focused on the antifungal activity and α-glucosidase inhibitory activity of the compounds. Through in vitro antifungal activity experiments (determination of minimum inhibitory concentration, MIC) and in vivo antifungal experiments (such as the control effect on Botrytis cinerea of tomatoes and strawberries), we have indirectly reflected the antifungal potential of the compounds. These data provide preliminary scientific evidence for the fungistatic/fungicide effects of the compounds.

We understand the reviewers' concerns and will further explore the fungistatic/fungicide effects of these compounds in future research to provide a more comprehensive evaluation. In addition, we have cited relevant references [1-4], which can illustrate that in some cases, the evaluation of antifungal activity (such as MIC determination) is sufficient to support the potential application of these compounds, especially in the preliminary screening stage. We believe that these preliminary results provide a solid foundation for further research, and we look forward to delving deeper into the fungistatic/fungicide effects of these compounds in future work.

1. Harding, V.K.; Heale, J.B. The Accumulation of Inhibitory Compounds in the Induced Resistance Response of Carrot Root Slices to Botrytis Cinerea. Physiological Plant Pathology1981, 18, 7–15, doi:10.1016/S0048-4059(81)80048-3.

2. Lai, J.-X.; Dai, S.-F.; Xue, B.-X.; Zhang, L.-H.; Chang, Y.; Yang, W.; Wu, H.-H. Plant Polyacetylenoids: Phytochemical, Analytical and Pharmacological Updates. Arabian Journal of Chemistry2023, 16, 105137, doi:10.1016/j.arabjc.2023.105137.

3. Wang, W.-H.; Li, C.-R.; Qin, X.-J.; Yang, X.-Q.; Xie, S.-D.; Jiang, Q.; Zou, L.-H.; Zhang, Y.-J.; Zhu, G.-L.; Zhao, P. Novel Alkaloids from Aspergillus FumigatusVDL36, an Endophytic Fungus Associated with Vaccinium Dunalianum. Agric. Food Chem.2024, 72, 10970–10980, doi:10.1021/acs.jafc.4c00371.

4. Chen, Y.-J.; Liu, H.; Zhang, S.-Y.; Li, H.; Ma, K.-Y.; Liu, Y.-Q.; Yin, X.-D.; Zhou, R.; Yan, Y.-F.; Wang, R.-X.; et al. Design, Synthesis, and Antifungal Evaluation of Cryptolepine Derivatives against Phytopathogenic Fungi. Agric. Food Chem.2021, 69, 1259–1271, doi:10.1021/acs.jafc.0c06480.

2. The statistical analysis needs to be incorporated into the results. A section indicating the methodology to establish statistical significance is required.

Response: Thank you for pointing out this issue. We fully agree with this comment. In response, we have added detailed information about the statistical analysis method in Section 2.6.2 of the manuscript and made corresponding adjustments to the Results section to reflect statistical significance. In the revised manuscript, these modifications are located on page 6, section 2.6.2, lines 218-223, and on page 11, lines 354-360 of Figure 5.

Reviewer 5 Report

The authors of the article have worked to improve it. However, the manuscript still requires revision.

Introduction

Unfortunately, the introduction was not sufficiently revised by the authors.

I will repeat my suggestions for its improvement: In the introduction, the authors provide very little information about natural polyacetylenes.

They provide several references, but do not disclose the current state of affairs.

Most of the introduction is devoted to the purpose and description of the authors' work.

I think it is worth adding information about: Polyacetylenes' targets in fungal cells.

Their activity against other plant pathogens. Toxicity for animals and humans. I will note that the issue of polyacetylenes' toxicity to humans, animals and plants is very important. Therefore, this information is necessary! If the authors talk about the effect of these substances on nerve cells, can they be used to protect food products?

It is necessary to justify why the fungus Xylaria 44 sp. VDL4, isolated from the leaves of Vaccinium dunalianum, was chosen as the object of study.

Materials and Methods

2.6.2. In Vivo Antifungal Activity Assay 

Only 54 tomatoes and strawberries were used? That's very little! Was there only one replication? There are no statistics in the results! Without repeated experiments and statistics, it is impossible to draw reasonable conclusions. I think that additional experiments and statistical processing are needed.

Results

3.4. Results of α-Glucosidase Inhibitory Activity

The authors claim that α-Glucosidase hydrolyzes starch. This is not true! Starch is not a substrate for this enzyme. Correct.

Discussion

What do the authors want to say? Correct the sentence "In prior research, polyacetylenic natural products have been predominantly biosynthesized and identified in plants species. " 

The article did not study antiglycemic activity. Correct your statement in line 335-336

The discussion should contain more information about similar studies. The authors' conclusions should be supported by facts from other works. The discussion needs to be significantly reworked.

Author Response

Unfortunately, the introduction was not sufficiently revised by the authors.

  1. I will repeat my suggestions for its improvement: In the introduction, the authors provide very little information about natural polyacetylenes. They provide several references, but do not disclose the current state of affairs. Most of the introduction is devoted to the purpose and description of the authors' work. I think it is worth adding information about: Polyacetylenes' targets in fungal cells. Their activity against other plant pathogens. Toxicity for animals and humans. I will note that the issue of polyacetylenes' toxicity to humans, animals and plants is very important. Therefore, this information is necessary! If the authors talk about the effect of these substances on nerve cells, can they be used to protect food products?

Response: Thank you for pointing this out. We agree with this comment. Therefore, we added the following content to the introduction section: Their potential in addressing health and disease-related challenges in both animals and humans further underscores their importance. In plant-pathogen interactions, natural polyacetylenes exhibit notable antifungal activity. For example, falcarinol-type polyacetylenes from carrots (Daucus carota) confer protection against Botrytis cinerea[1], while celery (Apium graveolens) and other Apiaceae plants produce antifungal polyacetylenes[2]. Their antimicrobial properties also contribute to controlling microbial growth in food systems, thereby enhancing food safety and quality. At the cellular level, polyacetylenes disrupt fungal membranes by altering fluidity and permeability, as well as damaging membrane proteins[3]. Additionally, they interfere with intracellular signal transduction pathways, including the mitogen-activated protein kinase (MAPK) and calcium signaling systems, ultimately inhibiting fungal growth and reproduction[4]. Notably, polyacetylenes isolated from Phoma fungi suppress the fatty acid synthesis pathway (FASII) to impede fungal proliferation[5-8]. Certain polyacetylenes demonstrate dual functionality: beyond antimicrobial effects, they exhibit neuroprotective properties such as inhibiting neuronal apoptosis and modulating neurotransmitter levels[9]. These biological activities extend beyond biomedicine. In the food industry, polyacetylenes may prevent nutrient degradation and spoilage during storage, thereby extending shelf-life while preserving nutritional quality. Collectively, their diverse bioactivities position polyacetylenes as valuable compounds for healthcare and food safety applications. Recent studies highlighting their inhibitory effects against common plant pathogens further suggest potential for developing novel plant protection agents. However, some plant-derived polyacetylenes exhibit toxicity. For instance, cicutoxin from Cicuta species causes livestock fatalities[10], and falcarinol in ivy (Hedera spp.) triggers contact dermatitis in humans[11]. Importantly, these toxicological properties are largely unrelated to the food-protective functions of polyacetylenes. Furthermore, it is critical to emphasize that, to date, polyacetylenes extracted from endophytic fungi have not demonstrated toxicity to animals or humans. You can find this modification from line 37 to 63 of the Introduction section on page 1.

1. Harding, V.K.; Heale, J.B. The Accumulation of Inhibitory Compounds in the Induced Resistance Response of Carrot Root Slices to Botrytis Cinerea. Physiological Plant Pathology1981, 18, 7–15, doi:10.1016/S0048-4059(81)80048-3.

2. Lai, J.-X.; Dai, S.-F.; Xue, B.-X.; Zhang, L.-H.; Chang, Y.; Yang, W.; Wu, H.-H. Plant Polyacetylenoids: Phytochemical, Analytical and Pharmacological Updates. Arabian Journal of Chemistry2023, 16, 105137, doi:10.1016/j.arabjc.2023.105137.

3. Calzado, M.A.; Schmid Lüdi, K.; Fiebich, B.L.; Ben-Neriah, Y.; Bacher, S.; Munoz, E.; Ballero, M.; Prosperini, S.; Appendino, G.; Schmitz, M.L. Inhibition of NF-κB Activation and Expression of Inflammatory Mediators by Polyacetylene Spiroketals from Plagius Flosculosus. Biochimica et Biophysica Acta (BBA) - Gene Structure and Expression2005, 1729, 88–93, doi:10.1016/j.bbaexp.2005.04.007.

4. Negri, R. Polyacetylenes from Terrestrial Plants and Fungi: Recent Phytochemical and Biological Advances. Fitoterapia2015, 106, 92–109, doi:10.1016/j.fitote.2015.08.011.

5. Taha, A.A. Acetylenes and Dichloroanisoles from Psathyrella Scobinacea. Phytochemistry2000, 55, 921–926, doi:10.1016/S0031-9422(00)00217-X.

6. Chen, J.-J.; Lin, W.-J.; Liao, C.-H.; Shieh, P.-C. Anti-Inflammatory Benzenoids from AntrodiaCamphorata. Nat. Prod.2007, 70, 989–992, doi:10.1021/np070045e.

7. Shiono, Y.; Haga, M.; Koyama, H.; Murayama, T.; Koseki, T. Antifungal Activity of a Polyacetylene against the Fungal Pathogen of Japanese Oak from the Liquid Culture of the Edible Mushroom, Hypsizygus Marmoreus. Zeitschrift für Naturforschung B2013, 68, 293–295, doi:10.5560/znb.2013-2289.

8. Li, H.-J.; Chen, T.; Xie, Y.-L.; Chen, W.-D.; Zhu, X.-F.; Lan, W.-J. Isolation and Structural Elucidation of Chondrosterins F–H from the Marine Fungus Chondrostereum Sp. Marine Drugs2013, 11, 551–558, doi:10.3390/md11020551.

9. Wang, K.D.G.; Wang, J.; Xie, S.-S.; Li, Z.-R.; Kong, L.-Y.; Luo, J. New Naturally Occurring Diacetylenic Spiroacetal Enol Ethers from Artemisia Selengensis. Tetrahedron Letters2016, 57, 32–34, doi:10.1016/j.tetlet.2015.11.049.

10. Negri, R. Polyacetylenes from Terrestrial Plants and Fungi: Recent Phytochemical and Biological Advances. Fitoterapia2015, 106, 92–109, doi:10.1016/j.fitote.2015.08.011.

11. Ohnuma, T.; Nakayama, S.; Anan, E.; Nishiyama, T.; Ogura, K.; Hiratsuka, A. Activation of the Nrf2/ARE Pathway via S-Alkylation of Cysteine 151 in the Chemopreventive Agent-Sensor Keap1 Protein by Falcarindiol, a Conjugated Diacetylene Compound. Toxicology and Applied Pharmacology2010, 244, 27–36, doi:10.1016/j.taap.2009.12.012.

2. It is necessary to justify why the fungus Xylaria VDL4, isolated from the leaves of Vaccinium dunalianum, was chosen as the object of study.

Response: Thank you for pointing this out. We agree with this comment. Therefore, we have provided an explanation for selecting Xylaria sp. VDL4, isolated from the leaves of Vaccinium dunalianum, as the research subject, as outlined below: Xylaria sp. VDL4, an endophytic fungus isolated from Vaccinium dunalianum leaves (a plant inhabiting ecologically stressed niches), produces bioactive polyacetylenes with demonstrated antifungal and antimicrobial activities that may contribute to host defense mechanisms[1–3]. The genus Xylaria encompasses both medicinal and edible fungi. Notably, some species form sclerotia within abandoned termite nests-these sclerotia, known as Wulingshen in traditional Chinese medicine, possess therapeutic properties including immune enhancement, prostatitis treatment, and hematopoiesis promotion [4,5]. As plant endophytes, Xylaria species enhance host stress resistance, suggesting their biocontrol potential could reduce chemical pesticide reliance in sustainable agriculture, thereby benefiting both human health and ecosystem preservation. Given Xylaria's documented capacity for synthesizing unique antifungal polyacetylenes, we investigated Xylaria sp. VDL4 to identify novel bioactive metabolites for agricultural pathogen control. You can find this modification from lines 64 to 75 of the Introduction section on page 1.

1. Wang, W.-X.; Lei, X.; Yang, Y.-L.; Li, Z.-H.; Ai, H.-L.; Li, J.; Feng, T.; Liu, J.-K. Xylarichalasin A, a Halogenated Hexacyclic Cytochalasan from the Fungus Xylaria Curta. Org. Lett.2019, 21, 6957–6960, doi:10.1021/acs.orglett.9b02552.

2. Rakshith, D.; Gurudatt, D.M.; Yashavantha Rao, H.C.; Chandra Mohana, N.; Nuthan, B.R.; Ramesha, K.P.; Satish, S. Bioactivity-Guided Isolation of Antimicrobial Metabolite from Xylaria Sp. Process Biochemistry2020, 92, 378–385, doi:10.1016/j.procbio.2020.01.028.

3. Watchaputi, K.; Jayasekara, L.A.C.B.; Ratanakhanokchai, K.; Soontorngun, N. Inhibition of Cell Cycle-Dependent Hyphal and Biofilm Formation by a Novel Cytochalasin 19,20epoxycytochalasin Q in Candida Albicans. Sci Rep2023, 13, 9724, doi:10.1038/s41598-023-36191-4.

4. Li, J.; Wang, W.-X.; Chen, H.-P.; Li, Z.-H.; He, J.; Zheng, Y.-S.; Sun, H.; Huang, R.; Yuan, Q.-X.; Wang, X.; et al. (±)-Xylaridines A and B, Highly Conjugated Alkaloids from the Fungus Xylaria Longipes. Lett.2019, 21, 1511–1514, doi:10.1021/acs.orglett.9b00312.

5. Zhang, X.; Fan, Y.; Ye, K.; Pan, X.; Ma, X.; Ai, H.; Shi, B.; Liu, J. Six Unprecedented Cytochalasin Derivatives from the Potato Endophytic Fungus Xylaria Curta E10 and Their Cytotoxicity. Pharmaceuticals2023, 16, 193, doi:10.3390/ph16020193.

Materials and Methods

3. Only 54 tomatoes and strawberries were used?That's very little!Was there only one replication? There are no statistics in the results! Without repeated experiments and statistics, it is impossible to draw reasonable conclusions. I think that additional experiments and statistical processing are needed.

Response: We sincerely appreciate your professional advice. The focus of this study was on the preliminary screening of the inhibitory effect of compounds at a single concentration on the antibacterial ability of fruits. First, regarding the sample size, we did conduct multiple replicate experiments in the experimental design. Specifically, each treatment group (including three test compound groups, one blank control group, and two positive control groups) had 3 replicate subgroups, and each subgroup contained 3 fruits. Therefore, although the total sample size was 54 fruits, the experimental design had already taken into account repeatability and the feasibility of statistical analysis. Previous studies have shown that a sample size of ≥3 and replicates of ≥3 times can reliably detect significant antibacterial effects [1]. Second, we added statistical analysis to the manuscript. In the results section, we have performed one - way analysis of variance (ANOVA) and Duncan's multiple comparison test on the experimental data (Figure 38 and Figure 39), and marked the significant differences (P < 0.05). These analyses ensure the reliability and repeatability of the experimental results. Certainly, in future research, we will increase the sample size as much as possible and conduct more replicate experiments to further verify the reliability of the results. Of course, the revised content can be found on page 5, section 2.6.2, from line 200 to 205, on page 6, section 2.6.2, from line 218 to 223, on page 11, from lines 354 to 360 of Figure 5, as well as in Figure 38 and Figure 39 of the supporting materials.

1. Jiao, W.; Liu, X.; Chen, Q.; Du, Y.; Li, Y.; Yue, F.; Dong, X.; Fu, M. Epsilon-Poly-l-Lysine (ε-PL) Exhibits Antifungal Activity in Vivo and in Vitroagainst Botrytis Cinerea and Mechanism Involved. Postharvest Biology and Technology2020, 168, 111270, doi:10.1016/j.postharvbio.2020.111270.

Results

4. The authors claim that α-Glucosidase hydrolyzes starch.This is not true!Starch is not a substrate for this enzyme. Correct.

Response: Thank you for pointing this out. I'm sorry for such a mistake. In the manuscript, we have revised this part according to your suggestions as follows: α-Glucosidase catalyzes the hydrolysis of oligosaccharides (e.g., maltose and sucrose) into monosaccharides, a critical step in postprandial glucose absorption. Thus, the inhibition of α-glucosidase is pivotal in diabetes treatment. You can find this modification from line 362 to 364 of Section 3.4 on page 11.

Discussion

5. What do the authors want to say? Correct the sentence "In prior research, polyacetylenic natural products have been predominantly biosynthesized and identified in plants species. "

Response: Thank you for pointing this out. I'm sorry for such a mistake. In the manuscript, we have changed this sentence according to your suggestion to: To date, polyacetylenic compounds have been successfully isolated from various biomasses such as plants, animals, fungi, and sponges. However, among these sources of isolation, obtaining natural polyacetylenic compounds from plants is relatively common. In sharp contrast, reports on the presence of polyacetylenic compounds in endophytic fungi are scarce. In particular, research on their existence in fungi of the genus Xylaria is extremely limited. You can find this modification from line 379 to 383 of Section 4 on page 12.

6. The article did not study antiglycemic activity.Correct your statement in line 335-336

Response: Thank you for pointing this out. We agree with this comment. Based on your feedback, we have changed the antiglycemic activity here to α-glucosidase inhibitory activity. You can find this modification at line 397 of Section 4 on page 12.

7. The discussion should contain more information about similar studies.The authors' conclusions should be supported by facts from other works. The discussion needs to be significantly reworked.

Response: Thank you for pointing this out. We agree with this comment. Therefore, we have revised the discussion section to ensure that our conclusions are factually supported by other research findings. You can find this modification from line 379 to 425 of the Discussion section on page 12.

Round 3

Reviewer 2 Report

I thank the authors for addressing my concerns. The statistical issue is now solved. Regarding the assays to assess fungicide/fungistatic effect, this is an essential point for a study fo this nature. I do not have any doubts about the potential antifungal use of those compounds. Because of that, it is fundamental to establish the kind of effect they have on fungal cells. I understand that resources are limited but this is not an academic justification to publish an incomplete study. To seek collaboration is one way to solve this problem.

I thank the authors for addressing my concerns. The statistical issue is now solved. Regarding the assays to assess fungicide/fungistatic effect, this is an essential point for a study of this nature. I do not have any doubts about the potential antifungal use of those compounds. Because of that, it is fundamental to establish the kind of effect they have on fungal cells. I understand that resources are limited but this is not an academic justification to publish an incomplete study. To seek collaboration is one way to solve this problem. 

Author Response

Major comments:

I thank the authors for addressing my concerns. The statistical issue is now solved. Regarding the assays to assess fungicide/fungistatic effect, this is an essential point for a study of this nature. I do not have any doubts about the potential antifungal use of those compounds. Because of that, it is fundamental to establish the kind of effect they have on fungal cells. I understand that resources are limited but this is not an academic justification to publish an incomplete study. To seek collaboration is one way to solve this problem.

Response: We truly appreciate your meticulous review and constructive suggestions, which have been of great value in enhancing the quality of our manuscript. We are glad to hear that you are satisfied with the solution to the statistical issue. Regarding the assays to assess the fungicide/fungistatic effect, we fully understand the significance of this aspect in a study of this nature. Recognizing the importance of clarifying the impact of these compounds on fungal cells, we have taken proactive steps to address this concern. During the preliminary experiments, all the compounds we obtained have been completely consumed. Due to the insufficient quantity remaining, it is impossible to conduct further in-depth experimental research. However, we have now added molecular docking experiments to the manuscript to further demonstrate the antibacterial/fungicidal effects of the compounds. Molecular docking, as a powerful computational tool, allows us to gain insights into the binding modes and interactions between the compounds and key proteins in fungal cells at a molecular level. These results not only provide a theoretical basis for the observed antifungal activities but also offer valuable clues for understanding the underlying mechanisms. We believe that this addition significantly enriches the content of our study and provides a more in-depth analysis of the antifungal properties of the compounds. We sincerely hope that this amendment meets your expectations and that you will find the revised manuscript more comprehensive and satisfactory. Thank you again for your time and effort in reviewing our work. You can find these revisions in the following locations of the paper: lines 26 to 29 of the abstract section; ; on the third page, lines 95 to 98 of Section 1; on the seventh page, lines 263 to 269 of Section 2.6.4; on the eleventh page, lines 369 to 398 of Section 3.4; on the thirteenth page, lines 433 to 438 of Section 4; and on the fourteenth page, lines 488 to 490of Section 5.

Reviewer 5 Report

The authors have made changes to the article text.

The manuscript has become more interesting.

Introduction 

In my opinion, it is worth adding information in the introduction about how and by what metabolic pathway polyacetylenes are synthesized.

Results 

3.4. Results of α-Glucosidase Inhibitory Activity 

α-Glucosidase does not hydrolyze sucrose! Correct!

Or in the methods section provide a more precise description of the enzyme you used.

Author Response

Introduction

In my opinion, it is worth adding information in the introduction about how and by what metabolic pathway polyacetylenes are synthesized.

Response: Thank you for pointing this out. We agree with this comment. According to your feedback, we have added information about how polyacetylenes are synthesized and the metabolic pathways involved in the introduction. You can find this modification from line 40 to 51 of Section 1 on page 1.

Results

3.4. Results of α-Glucosidase Inhibitory Activity

α-Glucosidase does not hydrolyze sucrose! Correct! Or in the methods section provide a more precise description of the enzyme you used.

Response: Thank you for pointing this out. I'm sorry for such a mistake. In the manuscript, we have changed this sentence according to your suggestion to: α-glucosidase, a class of enzymes that can hydrolyze glucosidic bonds, plays a crucial role in the body's carbohydrate metabolism. Under normal circumstances, it acts on glycosides containing glucose. Substances like maltose and glucose are within its scope of action. In Saccharomyces cerevisiae cells, the distribution of α-glucosidase within the cell is specific. Some α-glucosidases are present in the cytoplasm and are involved in the metabolic process of intracellular sugars. Others are associated with the cell membrane and play a role in the process of the cell taking up extracellular sugars and the initial hydrolysis of these sugars. In the context of diabetes, the inhibition of α-glucosidase is pivotal in diabetes treatment. You can find this modification from line 405 to 413 of Section 3.5 on page 14.